# SecureDL: Byzantine-Robust and Privacy-Preserving Aggregation for Decentralized Learning

## Abstract

In fully decentralized machine learning (DL), Byzantine failures remain a fundamental challenge. Though Byzantine-robust aggregation schemes have strengthened the resilience of DL against malicious updates, the decentralized nature of these systems, which requires clients to access each other's model updates, makes them more vulnerable to inference attacks than traditional federated learning. This reveals a fundamental tension between robustness and privacy, highlighting the need for aggregation schemes that can detect and neutralize malicious behavior while simultaneously preserving the confidentiality of model updates.

In this paper, we introduce **SecureDL**, to the best of our knowledge, the first decentralized learning protocol secure against Byzantine participants and simultaneously providing privacy guarantees. **SecureDL** leverages secret sharing-based multi-party computation to enhance the security and privacy of decentralized learning systems in adversarial settings, notably requiring only two semi-honest and non-colluding clients to guarantee privacy-preserving, robust training regardless of the total number of Byzantine participants. The protocol employs direction and magnitude consistency checks to filter adversarial model updates and uses scoring-based aggregation to achieve fair and resilient global models. Extensive evaluations across diverse datasets, under both IID and non-IID conditions, show that **SecureDL** consistently outperforms multiple state-of-the-art defenses, preserving both accuracy and convergence even under strong data and model poisoning attacks, including sophisticated adaptive threats.

## 1 Introduction

Collaborative machine learning is a paradigm shift enabling diverse entities to collaboratively train models with improved accuracy and robustness, while maintaining privacy of collaborators. This approach is particularly beneficial when data centralization is impractical due to privacy, bandwidth, or regulatory constraints. Arguably, the most well-known approach of collaborative machine learning is McMahan et al. (2017)'s Federated Learning (FL). FL features a central server which aggregates the individual model updates obtained from collaborators who train their models locally using their own private dataset. However, FL places an inherent trust in the central server, a potential single point of failure (Lian et al., 2017; Bonawitz et al., 2019; Ghavamipour et al., 2023), that may result in overlooking potential issues related to security and computational efficiency.

Decentralized Learning (DL) has gained popularity as a scalable and communication-efficient alternative to existing collaborative learning paradigms. Compared to FL schemes, Lian et al. (2017) show that DL eliminates the need for a central aggregator, which in turn enhances clients' privacy and prevents issues related to a single point of failure (Cheng et al., 2019; Vogels et al., 2021; Hu et al., 2019). Specifically, DL implements global model training by aggregating the model parameters on-device with peer-to-peer message exchanges between clients (similar to gossip-based algorithms over random networks (Hegedűs et al., 2019)), instead of centralized aggregation. In DL, every client actively updates its own model by using local data and updates received from its neighbors, resulting in optimized learning in terms of model accuracy and convergence speed. Given all these benefits, DL systems are particularly susceptible to privacy attacks, both under passive and active security models, with distinct privacy challenges compared to those encountered in FL (Pasquini et al., 2023).

Table 1: Comparison of our protocol **SecureDL** with existing robust aggregation schemes and their main defence mechanisms. Abbreviations: **EucD**: Euclidean Distance, **PerC**: Performance Check, **CosS**: Cosine Similarity, **Coord**: Coordinate-wise Operations, **Norm**: Normalization.

| Rules | Main defence mechanisms | | | | | Byzantine tolerance | Privacy |
|---|---|---|---|---|---|---|---|
| | EucD | PerC | CosS | Coord | Norm | | |
| DKrum | ✓ | | | | | $B < n/2$ | No |
| BRIDGE | | | | ✓ | | $B < n/2$ | No |
| UBAR | ✓ | ✓ | | | | $B \leq n - 1$ | No |
| DMedian | | | | ✓ | | $B < n/2$ | No |
| RAGD | ✓ | | | | ✓ | $B < n/2$ | No |
| BALANCE | ✓ | | | | ✓ | $B < n/2$ | No |
| DP-SGD | | | | ✓ | | $B < n/2$ | Heuristic |
| **SecureDL** | ✓ | | ✓ | | ✓ | $B \leq n - 2$ | Yes |

More specifically, the architecture of DL broadens the attack surface, allowing any client within the system to perform privacy attacks or cause additional data leakage.

In both FL and DL, the collaborative nature of these systems exposes them to significant risks from Byzantine adversaries (Lamport et al., 2019). These adversaries can exploit the system by compromising genuine clients, thereby submitting updates that undermine the model's integrity. Specifically, they can launch poisoning attacks in the form of data poisoning, where the local training data are tainted (Biggio et al., 2012), or model poisoning, where the updates sent to the central model are altered (Fang et al., 2020; Bhagoji et al., 2019). Such manipulations can drastically affect the global model's accuracy. This vulnerability highlights the pressing need for enhanced security protocols to protect the collaborative learning process from adversarial threats.

Numerous techniques have been developed to prevent poisoning attacks (mainly for FL) including coordinate-wise median (Yin et al., 2018), geometric median (Chen et al., 2017), trimmed-mean (Xie et al., 2018), and Krum (Blanchard et al., 2017) as part of the defences against both passive and active attackers, that are also applicable to the decentralized learning context (Fang et al., 2019; Yang and Bajwa, 2019; Peng et al., 2021; He et al., 2022; Mao et al., 2024; Fang et al., 2024). Despite the progress, a limitation of these methods is their reliance on clients receiving model updates directly from their collaborators, and thus observing the models of (or be observed by) other clients which compromises privacy if certain security measures are not in place. This consequently breaches a core tenant of DL (Pasquini et al., 2023) that aims to provide privacy protection for each collaborator's model without the need of an aggregator. Finally, Ye et al. (2024) investigate the trade-off between differential privacy and Byzantine robustness, which we refer as DP-SGD. They use coordinate-wise trimmed mean as the aggregation rule and study its robustness when Gaussian noise is injected into local updates. The work provides only heuristic privacy guarantees based on Renyi differential privacy. Table 1 summarizes these robust aggregation methods with their security and privacy guarantees and their key defense mechanisms. Further details on related work are provided in Appendix A.

In this paper, we introduce **SecureDL**, a privacy-preserving protocol for decentralized model training that, to the best of our knowledge, is the first to achieve resilience against Byzantine attacks. By employing a defence mechanism based on cosine similarity, normalization, and Euclidean distance, combined with secret sharing, the protocol enables clients to collaboratively detect malicious updates while keeping their own model updates private. The core of **SecureDL** is its Byzantine-resilient aggregation process, in which each received update is carefully evaluated with respect to its direction relative to the reference model. Updates that do not meet a similarity threshold are securely filtered out, while accepted ones are weighted according to their consistency with the reference. To ensure coherence, the framework normalizes client updates to a common scale and dynamically weights their influence in aggregation, giving greater emphasis to updates that are closer in magnitude to the reference model. These functionalities are supported by secure implementations of comparison, inversion, square root, and normalization operations in addition to the basic arithmetic over secret-shared data. Finally, **SecureDL** guarantees that each client only observes the aggregated outcome of its neighbors' updates, thereby mitigating the privacy attack surface in decentralized learning even under a dishonest majority.

Thus, the main contributions of the paper are as follows:

- We introduce **SecureDL**, a novel privacy-preserving and Byzantine-resilient decentralized learning method that utilizes secure multi-party computation (MPC). To the best of our knowledge, it is the first protocol achieving both privacy and Byzantine resilience in decentralized setting.
- We provide theoretical analysis regarding privacy guarantees offered by **SecureDL** and the impact of the changes we made on the model convergence of the overall training process.
- We empirically compare the resilience of **SecureDL** against a suite of state-of-the-art Byzantine attacks with the existing defences over MNIST, Fashion-MNIST, SVHN and CIFAR-10 datasets.
- We perform a thorough empirical analysis to assess overheads imposed by our secure aggregation protocol, to clearly characterize the cost of privacy protection on system performance in terms of computational time and network communications.

## 2 PRELIMINARIES

In this section, we provide the necessary background on decentralized learning and introduce the secure building blocks used in this paper.

### 2.1 LEARNING IN A DECENTRALIZED SETTING

In a decentralized learning system, each client $c$ from the group $C$ directly communicates with a selected group known as their neighbors $\mathbf{N}(c)$. These connections can be either static, formed at the beginning, or dynamic, changing over time. The network of clients forms an undirected graph $G = \{C, \cup_{c \in C} \mathbf{N}(c)\}$, where the vertices represent clients and the edges indicate the connections between them (Lian et al., 2017).

In a formal setting, client $c$ in the set $C$ owns a private dataset $D_i = \{(x_i, y_i)\}_i$ drawn from a hidden distribution $\xi_i$ and combining the private datasets would produce the global dataset $D$ with distribution $\xi$. Each client begins with a shared set of model parameters denoted as $w^0$. The aim of the training process is to identify the optimal parameters, denoted as $\mathbf{w}^*$, for a machine learning model. These parameters are sought to minimize the expected loss across the entire global dataset $D$:

$$\mathbf{w}^* = \arg \min_{\theta} \frac{1}{|\mathbf{N}(c)|} \sum_{n_i \in \mathbf{N}(c)} \underbrace{\mathbb{E}_{s_i \sim D_i} \left[ \mathcal{L}(\theta; s_i) \right]}_{\mathcal{L}_i} \tag{1}$$

where for each client $n_i$ in the network, $\mathbb{E}_{s_i \sim D_i} \left[ \mathcal{L}(\theta; s_i) \right]$ calculates the expected loss for that client's dataset $D_i$, where $s_i$ is a sample from $D_i$. Thus, the formula aims to find the model parameters $\theta$ that minimize the average expected loss across all clients in the network.

This process continues through several stages until reaching a predetermined stopping point. The stages are as follows: First, at step $t$, each client $c$ carries out gradient descent on their model parameters, leading to an interim model update, symbolized as $w_c^{t+1/2}$. Then, the clients exchange their interim model updates $w_c^{t+1/2}$ with their neighbors $\mathbf{N}(c)$. Concurrently, they also acquire the updates from these neighbors. Finally, the clients merge the updates received from their neighbors with their own, and use this combined update to modify their local state. A basic approach to aggregation involves averaging all the updates received. Mathematically, this is represented as $w^{t+1} = \frac{1}{|\mathbf{N}(c)|} \sum_{c \in \mathbf{N}(c)} w_c^{t+1/2}$. The DL training procedure is given in Algorithm 2 in Appendix I.

### 2.2 SECRET SHARING

We implement the authenticated additive secret sharing protocol given in Cramer et al. (2018) over the base rings $\mathbb{Z}_{2^k}$. For a security parameter $s$, each participant $P_i$ obtains a share $[\alpha]_i \in \mathbb{Z}_{2^s}$, ensuring $\alpha \equiv_{k+s} \sum_i [\alpha]_i$, where $\alpha \in \mathbb{Z}_{2^{k+s}}$ acts as the secret MAC (Message Authentication Code) key. For any secret $x \in \mathbb{Z}_{2^k}$, $P_i$ receives an additive share $[x]_i \in \mathbb{Z}_{2^{k+s}}$, satisfying the relation $x \equiv_k \sum_i [x]_i$.

Each participant $P_i$ broadcasts their share $[x]_i$ to collaboratively reconstruct the secret $x$. This scheme exhibits a full threshold characteristic; that is, the possession of all but one of the shares does not compromise the secret's confidentiality. Additionally, each $P_i$ is allocated a share $[\gamma_x]_i \in \mathbb{Z}_{2^{k+s}}$, fulfilling $\sum_i [\gamma_x]_i \equiv k + s\alpha \sum_i [x]_i$. Here, $\gamma_x \in \mathbb{Z}_{2^{k+s}}$ functions as the MAC of $x$. These MACs are

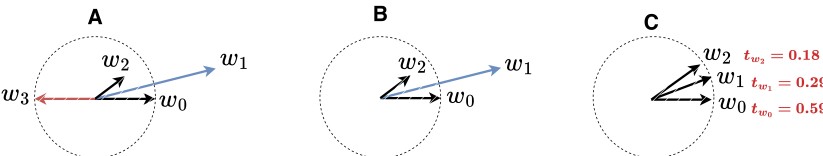

Figure 1: The illustration of **SecureDL** aggregation rule. The model update of client $c_0$, denoted as $w_0$, obtains the updates from other clients ($w_1$, $w_2$, and $w_3$). In step **A**, the protocol discards any model update with a negative cosine value. Step **B** depicts $c_0$ normalizing the received models based on its own model's magnitude. Step **C** shows the final accepted and normalized model updates, where **SecureDL** also assigns a trust score to each update based on its distance to the receiver's model.

pivotal in the MAC check protocol, validating the integrity of the shared secrets and identifying any malevolent activities. Each pair $[\![x]\!]_i = ([x]_i, [\gamma_x]_i)$ represents an authenticated share of $x$ held by $P_i$. The scheme permits local arithmetic on these authenticated shares. The implementation details of fundamental operations on secret-shared values, including arithmetic operations, secure comparison, inversion, and square root computation, are provided in Appendix B.1.

## 3 SecureDL

This section explains the secure aggregation rule within **SecureDL** in the presence of malicious clients. As the threat model, we consider the potential risks posed by participating clients under the assumption of a dishonest majority. We focus on malicious clients who intentionally deviate from the established procedures, engaging in disruptive or dishonest actions. These actions may include manipulating outcomes, injecting false data, or completely hindering system operations, posing significant threats to the system's integrity and reliability. Details on the threat model and problem statement can be found in Appendix C.

Byzantine clients manipulate model update directions, steering the global model away from its intended trajectory. This strategic deviation hinders the model's convergence. The resulting angular disparity between the model updates from these malicious clients and the global model of the receivers of those updates exceeds that of benign clients. In **SecureDL**, the direction of received model updates is evaluated against the global model of the receiving client using the **cosine similarity** metric (as illustrated in Figure 1). This metric is widely used to compute the angular difference between two vectors. The cosine similarity between two vectors $\mathbf{w}_a$ and $\mathbf{w}_b$ is expressed as:

$$\text{Cosine}(\mathbf{w}_a, \mathbf{w}_b) = \frac{\mathbf{w}_a \cdot \mathbf{w}_b}{\|\mathbf{w}_a\| \|\mathbf{w}_b\|}.$$

In our protocol, we address the challenge of computing cosine similarity in a decentralized learning environment under strong privacy guarantees. Clients only have access to aggregated information and not individual model updates, making computing similarity securely more challenging. To overcome this, we leverage collaborative client efforts to perform secure vectorized operations, supported by specialized protocols for computing square roots and inversions in a privacy-preserving manner. These secure operations enable the calculation of cosine similarities between secret-shared model update vectors $[\![\mathbf{w}_a]\!]$ and $[\![\mathbf{w}_b]\!]$, without revealing the underlying model updates.

The computation of cosine similarity within our secure framework involves several tightly integrated steps, as detailed in Algorithm 5. First, we apply the Beaver multiplication protocol to securely compute the dot product between two model updates without revealing their contents. To compute the norm of each model, we securely square the vector components, sum the results, and apply a square root operation, following the method described in Section B.4. After obtaining the norms, we use the inversion protocol from Section B.3 to compute the inverse of the product of these two norms. These secure operations, including dot product computation, norm calculation, and inversion, enable the privacy-preserving calculation of cosine similarity, ensuring that no sensitive information is leaked during the computation process.

Once the cosine similarities are computed, the next step is determining which model updates should be accepted or rejected based on a similarity threshold. Since the similarity scores remain in a secret-

---

**Algorithm 1: SecureDL**

---

**Input** : $n$ clients $C_0, C_1, \ldots, C_{n-1}$ with local training datasets $D_i$ for $i = 0, 1, \ldots, n-1$;
Number of global iterations $R_g$; Cosine similarity threshold $0 < \tau < 1$;

**Output :** Global model $[\![\mathbf{w}]\!]$ trained on all clients' datasets for each client

1 **Initialization step:**

2 All clients start from the same initial model $\mathbf{w}^0$.

3 **for** $r = 0, 1, 2, \ldots, R_g$ **do**

4    **Local training step:**

5    **for** *each client* $i, j \in \{0, 1, \ldots, n-1\}$ **do**

6       $\mathbf{w}_i \leftarrow \text{LocalUpdate}(\mathbf{w}^r, D_i, \beta)$

7       Generates shares $\{[\![\mathbf{w}_i]\!]_j\}_{j=0}^{n-1}$ from $\mathbf{w}_i$ and send $[\![\mathbf{w}_i]\!]_j$ to client $C_j$

8       Receives $\{[\![\mathbf{w}_j]\!]_i\}_{j=0}^{n-1}$

9    **Aggregation step:**

10    **for** *each client* $i, j \in \{0, 1, \ldots, n-1\}$ **do**

11       Set $[\![\mathbf{w}_i]\!]$ as the reference model

12       Initialize trust scores $[\![s_{ij}]\!]$

13       **if** $i \neq j$ **then**

14          Compute cosine similarity securely: $[\![\cos(\mathbf{w}_i, \mathbf{w}_j)]\!] = \dfrac{[\![\langle [\![\mathbf{w}_i]\!], [\![\mathbf{w}_j]\!]\rangle]\!]}{[\![\|[\![\mathbf{w}_i]\!]\|]\!] \cdot [\![\|[\![\mathbf{w}_j]\!]\|]\!]}$

15          **if** SECURECOMPARE$([\![\cos(\mathbf{w}_i, \mathbf{w}_j)]\!], \tau)$ **then**

16             Normalize: $[\![\mathbf{w}'_j]\!] \leftarrow [\![\mathbf{w}_j]\!] \cdot \left([\![\|\mathbf{w}_i\|]\!] \cdot [\![\|\mathbf{w}_j\|]\!]^{-1}]\!]\right)$

17             Compute secure squared distance: $[\![d_{ij}^2]\!] \leftarrow \left\langle [\![\mathbf{w}_i - \mathbf{w}'_j]\!], [\![\mathbf{w}_i - \mathbf{w}'_j]\!]\right\rangle$

18             Compute trust score : $[\![s_{ij}]\!] \leftarrow \dfrac{1}{[\![d_{ij}^2]\!] + \epsilon}$

19          **else**

20             Set $[\![s_{ij}]\!] = 0$

21       Normalize trust scores: $[\![\tilde{s}_{ij}]\!] = \dfrac{[\![s_{ij}]\!]}{\sum_{k=0}^{n-1}[\![s_{ik}]\!]}$

22       Aggregate weighted model updates: $[\![\hat{\mathbf{w}}_i]\!] = \sum_{j=0}^{n-1}[\![\tilde{s}_{ij}]\!] \cdot [\![\mathbf{w}'_j]\!]$

23       Reveal the new aggregated model update share: $Reveal([\![\hat{\mathbf{w}}_i]\!])$

24       Locally compute the new global model: $\mathbf{w}^{r+1} \leftarrow \sum_{j=0}^{n-1}[\![\hat{\mathbf{w}}_i]\!]_j$

---

shared form, direct comparison with the threshold is not possible without compromising privacy. We employ a secure comparison protocol described in Section B.2 to perform this comparison. This protocol allows clients to compare secret-shared cosine similarity scores against a predefined threshold without revealing their actual values. By securely filtering out updates that do not meet the threshold, **SecureDL** enhances the decentralized learning process's robustness while preserving the model updates' privacy and integrity.

Model updates that pass the direction similarity threshold are normalized to prevent Byzantine adversaries from manipulating their magnitudes. Compromised clients may attempt to submit model updates with exaggerated magnitudes, disproportionately influencing the receiver's global model update. To counter this risk, **SecureDL** normalizes each accepted model update to match the magnitude of the receiver's model using **L2 normalization**. The normalization of a secret-shared model update $[\![\mathbf{w}_a]\!]$ with respect to a receiver's model $[\![\mathbf{w}_b]\!]$ is computed as:

$$\text{Norm}([\![\mathbf{w}_a]\!], [\![\mathbf{w}_b]\!]) = [\![\mathbf{w}_a]\!] \times \frac{\|[\![\mathbf{w}_b]\!]\|_2}{\|[\![\mathbf{w}_a]\!]\|_2}.$$

The normalization procedure is securely performed as detailed in Algorithm 3. This adjustment ensures that each model update contributes proportionally during aggregation, reducing the influence of adversarially scaled updates and strengthening the robustness of the learning process.

**SecureDL** assigns a trust score to each accepted model update with a similar direction and magnitude to the receiver's model, instead of treating all updates equally. This trust score is based on the

**Euclidean distance** between the secret-shared normalized model update $[\![\mathbf{w}'_j]\!]$ and the receiver's model $[\![\mathbf{w}_i]\!]$, and is computed securely using the procedure detailed in Algorithm 4. The trust score is based on the **Euclidean distance** between the secret-shared normalized model update and the receiver's model update. This distance is securely computed within our protocol, and the trust score is then assigned as the inverse of the distance, with a small constant $\epsilon$ added for numerical stability:

$$[\![s_{ij}]\!] = \frac{1}{[\![d_{ij}]\!] + \epsilon}.$$

The computed trust scores for accepted updates are normalized to ensure they sum to one prior to aggregation. This distance-based scoring ensures that updates closer to the receiver's model have greater influence during aggregation, while more distant — and potentially malicious — updates are assigned less weight, while preserving the privacy of individual updates.

The complete **SecureDL** protocol, including the integrated trust scoring system, is outlined in Algorithm 1. At each global iteration, clients accept model updates that pass the similarity threshold, normalize them to align magnitudes, compute trust scores based on Euclidean distance, normalize the trust scores, and aggregate the updates in a weighted manner accordingly. This process is repeated over multiple global iterations to complete the training protocol. Through this design, **SecureDL** achieves robust, privacy-preserving aggregation while effectively mitigating the impact of Byzantine adversaries on decentralized model training.

### 3.1 Convergence

We demonstrate that **SecureDL** achieves convergence even in the presence of arbitrary Byzantine clients. Specifically, for an arbitrary number of malicious updates, the difference between the model learned by **SecureDL** and the optimal model under no attack remains provably bounded. Our proof builds on Cao et al. (2020), adapted to the trust-scored weighted aggregation mechanism in **SecureDL**. Full assumptions and the complete proof are provided in Appendix G.

**Theorem 1** (Convergence). *Suppose the assumptions in Appendix G hold, and **SecureDL** uses $R_g = 1$ and $\beta = 1$. Then, for any number of malicious clients, the distance between the global model $\mathbf{w}^t$ at iteration $t$ and the optimal model $\mathbf{w}^*$ satisfies, with probability at least $1 - \delta$,*

$$\|\mathbf{w}^t - \mathbf{w}^*\| \le (1 - \rho)^t \|\mathbf{w}^0 - \mathbf{w}^*\| + \frac{12\alpha\Delta_1}{\rho},$$

*where $\alpha$ is the global learning rate, $\rho$ and $\Delta_1$ are positive constants from Cao et al. (2020), and the normalized trust scores satisfy $\sum_{j \in \mathcal{S}} \tilde{s}_{ij} = 1$ for each client $i$. Moreover, when $|1 - \rho| < 1$, as $t \to \infty$, $\lim_{t \to \infty} \|\mathbf{w}^t - \mathbf{w}^*\| \le \frac{12\alpha\Delta_1}{\rho}$, thus confirming convergence.*

### 3.2 Privacy Analysis

We demonstrate that **SecureDL** achieves privacy-preserving aggregation against malicious adversaries using a simulation-based proof technique (Goldreich, 1998; Lindell, 2017). In simulation-based security, privacy is guaranteed by showing the existence of a polynomial-time simulator that can replicate the adversary's view without access to honest parties' private inputs. This ensures that adversaries cannot distinguish between the real execution and a simulated one, thereby learning nothing about the honest clients' private information. Below, we present an informal statement of the main theorem. A detailed simulation-based proof is provided in Appendix H.

**Theorem 2** (Privacy). *Consider an $n$-party secure multi-party computation protocol $\Pi$, designed to compute a function $f$ over inputs $\bar{x} := (x_1, \ldots, x_n)$ using the **SecureDL** algorithm. Each party $P_1, P_2, \ldots, P_n$ holds an authenticated additive share of their private inputs. For the protocol $\Pi$ to be considered secure against malicious adversaries, it must satisfy the following condition: for every subset of parties $I \subseteq [n]$, there exists a probabilistic polynomial-time simulator $S_I$ such that:*

$$\{S_I(\bar{x}, f_I(\bar{x}))\}_{\bar{x}} \overset{comp}{\equiv} \{\mathrm{REAL}_\Pi(\bar{x})\}_{\bar{x}},$$

*where $f_I(\bar{x})$ denotes the outputs relevant to the honest subset $I$, and $\mathrm{REAL}_\Pi(\bar{x})$ represents the adversary's view during the real execution of $\Pi$.*

# 4 EVALUATION

We conducted a series of experiments to evaluate the robustness of **SecureDL** in the presence of Byzantine adversaries and to measure the overhead introduced by the proposed modifications. For empirical comparison, we also implemented Median and Krum (referred to as 'DMedian' and 'DKrum'), along with UBAR and BRIDGE, to benchmark our protocol under various Byzantine scenarios in decentralized learning. Additional experimental results and analysis, including robustness under non-iid conditions, performance overhead, and complexity evaluation, are provided in Appendix F.

## 4.1 EXPERIMENTAL SETUP

We evaluate **SecureDL** using PyTorch (Paszke et al., 2019) on a high-performance computing cluster equipped with modern GPUs and multi-core CPUs. Our experiments cover four popular deep learning datasets: MNIST (LeCun, 1998), Fashion-MNIST (Xiao et al., 2017), SVHN (Netzer et al., 2011), and CIFAR-10 (Krizhevsky et al., 2009). We partition the datasets either i.i.d. or non-i.i.d. among clients. For model training, we employ a lightweight Multi-Layer Perceptron (MLP) for grayscale datasets (MNIST, Fashion-MNIST) and a Convolutional Neural Network (CNN) for color datasets (CIFAR-10, SVHN). A detailed description of the datasets, model architectures, data partitioning, and system environment is provided in Appendix D.

## 4.2 ATTACKS ON AGGREGATION SCHEMES

In this section, we compare **SecureDL** with a range of decentralized robust aggregation schemes, as summarized in Table 2, across the CIFAR-10, SVHN, and Fashion-MNIST datasets. We focus on accuracy under a variety of Byzantine attacks to evaluate both the resilience and limitations of these approaches. For the SVHN experiments, we employed a more advanced CNN model architecture to better capture the complexity of the dataset and ensure a fair comparison among all methods. In our experiments, we evaluate Sign-Flipping (SF), Gaussian Noise (Noise), Scaling Attack (SA), Label-Flipping (LF), Combination Attack (Combi), and an Adaptive Attack (Adapt). The detailed descriptions of these attacks are provided in Appendix E.

On the CIFAR-10 dataset (Table 2a), the *mean* method is vulnerable to all attack types, with accuracy dropping drastically in all adversarial settings, and reaching its lowest values under the combined and adaptive attacks. Among robust schemes, DKrum shows moderate resilience but still loses substantial accuracy under stronger attacks, such as 73.32% under noise and 78.54% under adaptive attack. In contrast, BRIDGE, UBAR, BALANCE, and RAGD deliver high accuracy across most scenarios; UBAR achieves 81.96% under scaling attacks, and BALANCE attains 82.64% under noise. However, under the most difficult attack types, including combined and adaptive, all these schemes experience some degree of accuracy degradation. Notably, **SecureDL** maintains relatively high and stable accuracy, with 76.09% under combined attacks and 78.05% under adaptive attacks, indicating strong resilience even when other robust schemes are more affected.

For the SVHN dataset (Table 2b), most robust aggregation rules—including BRIDGE, UBAR, BALANCE, and RAGD—demonstrate excellent performance, maintaining accuracy above 94% across nearly all attack types; for instance, BRIDGE achieves 95.34% and UBAR achieves 95.57% under sign-flipping attacks. Notably, all robust methods, including **SecureDL**, experience only a minor reduction in accuracy across different attack types, with performance typically decreasing by less than 2% relative to the attack-free baseline. This limited reduction may be due to the relative

Table 2: Accuracy comparison of existing decentralized robust aggregation schemes against **SecureDL** under various attacks over SVHN, CIFAR-10, and Fashion-MNIST datasets.

(a) CIFAR-10 dataset

| # Client / # Byzantine | N = 10 | | | | | | |
| Attack type | 0 | 2 | 2 | 2 | 2 | 4 | 4 |
| --- | --- | --- | --- | --- | --- | --- | --- |
| | W/O | SF | SA | Noise | LF | Combi | Adapt |
| Mean | 84.21 | 53.83 | 13.43 | 16.32 | 60.27 | 10.16 | 10.21 |
| DKrum | 82.20 | 73.55 | 74.30 | 73.32 | 72.95 | 73.52 | 78.54 |
| BRIDGE | 83.28 | 81.86 | 81.57 | 81.26 | 81.34 | 80.07 | 77.87 |
| UBAR | 83.36 | 82.06 | 81.96 | 81.64 | 82.56 | 79.45 | 78.21 |
| DMedian | 83.52 | 80.36 | 77.64 | 82.64 | 80.29 | 80.75 | 77.92 |
| RAGD | 83.06 | 79.28 | 75.51 | 79.28 | 78.83 | 77.29 | 77.29 |
| BALANCE | 83.02 | 77.38 | 82.64 | 80.38 | 72.69 | 73.83 | 78.12 |
| **SecureDL** | 83.69 | 81.99 | 81.33 | 81.55 | 76.11 | 76.09 | 78.05 |

(b) SVHN dataset

| # Client / # Byzantine | N = 10 | | | | | | |
| Attack type | 0 | 2 | 2 | 2 | 2 | 4 | 4 |
| --- | --- | --- | --- | --- | --- | --- | --- |
| | W/O | SF | SA | Noise | LF | Combi | Adapt |
| Mean | 95.65 | 21.34 | 22.39 | 19.99 | 94.82 | 55.35 | 10.32 |
| DKrum | 93.57 | 93.67 | 93.43 | 93.71 | 93.80 | 93.84 | 93.24 |
| BRIDGE | 95.45 | 95.34 | 95.34 | 95.37 | 95.38 | 94.98 | 94.21 |
| UBAR | 95.46 | 95.57 | 95.39 | 95.40 | 95.44 | 94.94 | 94.27 |
| DMedian | 95.24 | 94.96 | 94.67 | 95.29 | 94.86 | 94.71 | 93.21 |
| RAGD | 95.05 | 94.79 | 92.69 | 94.62 | 94.77 | 94.20 | 93.45 |
| BALANCE | 95.24 | 94.41 | 95.10 | 94.59 | 93.29 | 93.37 | 95.21 |
| **SecureDL** | 94.64 | 94.60 | 94.65 | 94.31 | 91.53 | 94.06 | 93.90 |

(c) Fashion-MNIST dataset

| # Client / # Byzantine | N = 10 | | | | | | |
| Attack type | 0 | 2 | 2 | 2 | 2 | 4 | 4 |
| --- | --- | --- | --- | --- | --- | --- | --- |
| | W/O | SF | SA | Noise | LF | Combi | Adapt |
| Mean | 93.28 | 78.12 | 25.23 | 19.14 | 87.04 | 21.13 | 13.22 |
| DKrum | 92.37 | 91.76 | 92.07 | 91.96 | 92.22 | 91.97 | 92.23 |
| BRIDGE | 93.24 | 93.04 | 92.99 | 92.90 | 92.79 | 92.57 | 92.12 |
| UBAR | 93.22 | 93.08 | 92.98 | 93.19 | 89.62 | 92.31 | 91.89 |
| DMedian | 93.02 | 92.46 | 92.73 | 92.89 | 92.49 | 92.71 | 92.22 |
| RAGD | 93.79 | 93.34 | 93.60 | 93.63 | 93.38 | 91.06 | 91.19 |
| BALANCE | 93.89 | 93.62 | 93.67 | 93.04 | 92.43 | 92.25 | 92.01 |
| **SecureDL** | 93.23 | 92.86 | 92.25 | 92.99 | 90.30 | 92.63 | 92.90 |

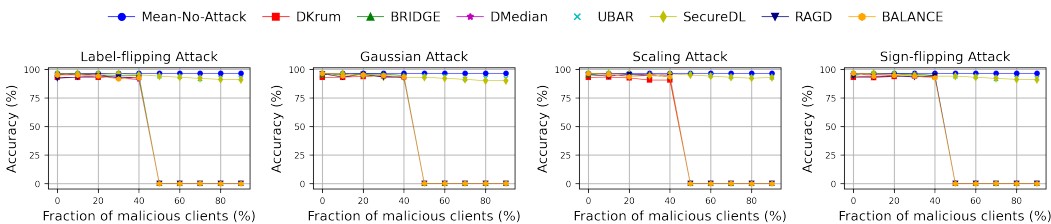

Figure 2: Varying fraction of malicious clients in different attack scenarios on MNIST dataset.

simplicity of the SVHN dataset combined with the capacity of the chosen model, which together make it more difficult for attacks to have a significant effect. **SecureDL** remains competitive, with accuracy consistently above 93%, closely tracking the top-performing methods and demonstrating that all leading robust approaches are effective on this dataset, even under the adaptive attack.

On the Fashion-MNIST dataset (Table 2c), the MEAN method again suffers drastic performance loss under attack, while DKrum and DMedian offer only moderate protection, with DKrum reaching 91.97% under combined attacks and DMedian attaining 92.71% in the same setting. In contrast, BRIDGE, UBAR, BALANCE, and RAGD deliver high accuracy across most scenarios; BRIDGE achieves 92.99% under scaling attacks, and RAGD attains 93.79% in the absence of attack. **SecureDL** consistently achieves among the highest accuracies, particularly under combined and adaptive attacks, with 92.63% and 92.90% respectively, matching or surpassing other robust schemes in these challenging settings. These results highlight both the effectiveness of **SecureDL** and the notable progress made by recent aggregation strategies in addressing a wide range of adversarial threats.

As shown in Table 2, the performance comparison across the CIFAR-10, SVHN, and Fashion-MNIST datasets demonstrates that **SecureDL** achieves consistently strong accuracy across diverse attack scenarios, performing comparably to other robust aggregation methods and outperforming them under certain attacks. These results support its suitability for decentralized learning scenarios that demand strong resilience against adversarial behavior while preserving model performance.

Figure 2 illustrates the performance of various aggregation schemes under four types of Byzantine attacks—Gaussian, Label-flipping, Sign-flipping, and Scaling—on the MNIST dataset, with the fraction of malicious clients varying from 0% to 80%. Both **SecureDL** and UBAR exhibit outstanding robustness, maintaining high accuracies even when up to 80% of the clients are malicious. In contrast, classical defenses such as DKrum, BRIDGE, and DMedian suffer substantial performance degradation once the fraction of adversaries approaches 50%, revealing their inherent Byzantine tolerance limits. BALANCE shows strong resilience when the proportion of malicious clients remains below 50%, sustaining competitive accuracies close to 94%, but its effectiveness diminishes beyond that threshold.

Under Scaling and Sign-flipping attacks, DKrum and BRIDGE witness severe drops in accuracy below 88% once the fraction of malicious clients exceeds 40%. DMedian performs moderately better, maintaining reasonable accuracy up to around 50% adversaries. RAGD offers slight improvements over DKrum and BRIDGE but remains vulnerable when the malicious fraction increases further. Although BALANCE retains notable robustness under these attacks for lower adversarial ratios, it cannot match the resilience of **SecureDL** and UBAR at higher attack levels. Under Gaussian and Label-flipping attacks, **SecureDL** and UBAR consistently dominate, preserving high accuracies across all levels of adversarial participation.

### 4.3 ROBUSTNESS UNDER NON-IID SETTING

To evaluate the robustness of **SecureDL** under realistic decentralized learning conditions, we conduct experiments using non-iid (non–independent and identically distributed) training data across clients. We simulate non-iid distributions on four datasets. In each experiment, a total of 10 clients participate, of which 4 act as Byzantine attackers. Various adversarial strategies, described in Appendix E, are applied. Different levels of non-iid data distribution are controlled through a skewness parameter, allowing us to systematically examine the impact of data heterogeneity. The resulting accuracy across different attack types and non-iid levels is summarized in Table 3, while a detailed description of the experimental settings and comprehensive analysis of the results are provided in Appendix F.1.

Table 3: Performance of **SecureDL** against different attacks and non-iid distributions, considering 4 Byzantine attackers among 10 clients.

(a) MNIST

| Attack | Level of Non-IID (Skewness) | | | | | |
|---|---|---|---|---|---|---|
| | 0 | 0.2 | 0.4 | 0.6 | 0.8 | 0.95 |
| W/O | 97.10 | 97.16 | 96.15 | 91.48 | 88.34 | 80.32 |
| SF | 96.79 | 96.85 | 96.08 | 91.13 | 89.34 | 80.30 |
| SA | 96.79 | 96.64 | 96.00 | 91.45 | 87.98 | 81.14 |
| Noise | 96.75 | 96.71 | 96.28 | 91.08 | 87.39 | 77.85 |
| LF | 96.70 | 96.58 | 95.43 | 90.72 | 87.73 | 78.73 |
| Combi | 96.57 | 96.44 | 95.18 | 90.78 | 89.54 | 81.12 |
| Adapt | 96.25 | 96.03 | 95.50 | 90.27 | 87.90 | 80.72 |

(b) VHDS

| Attack | Level of Non-IID (Skewness) | | | | | |
|---|---|---|---|---|---|---|
| | 0 | 0.2 | 0.4 | 0.6 | 0.8 | 0.95 |
| W/O | 94.64 | 94.58 | 94.36 | 93.92 | 92.85 | 89.55 |
| SF | 94.60 | 94.51 | 94.04 | 93.62 | 92.19 | 87.26 |
| SA | 94.65 | 94.53 | 94.15 | 93.85 | 92.47 | 88.25 |
| Noise | 94.31 | 94.24 | 94.00 | 93.44 | 91.99 | 86.60 |
| LF | 91.53 | 90.80 | 90.41 | 90.49 | 90.90 | 88.53 |
| Combi | 93.67 | 93.53 | 93.25 | 92.37 | 91.10 | 84.83 |
| Adapt | 93.90 | 93.58 | 93.39 | 92.83 | 91.62 | 86.90 |

(c) Fashion-MNIST

| Attack | Level of Non-IID (Skewness) | | | | | |
|---|---|---|---|---|---|---|
| | 0 | 0.2 | 0.4 | 0.6 | 0.8 | 0.95 |
| W/O | 87.25 | 87.35 | 87.24 | 87.37 | 87.36 | 87.34 |
| SF | 87.18 | 87.23 | 87.03 | 87.16 | 87.27 | 87.22 |
| SA | 87.25 | 87.34 | 87.40 | 87.21 | 87.27 | 87.42 |
| Noise | 87.28 | 87.31 | 87.25 | 87.37 | 87.36 | 87.33 |
| LF | 87.18 | 87.63 | 87.39 | 87.48 | 87.21 | 87.01 |
| Combi | 86.82 | 86.79 | 86.69 | 86.80 | 86.65 | 86.56 |
| Adapt | 86.79 | 86.73 | 86.66 | 86.63 | 86.70 | 86.50 |

(d) CIFAR-10

| Attack | Level of Non-IID (Skewness) | | | | | |
|---|---|---|---|---|---|---|
| | 0 | 0.2 | 0.4 | 0.6 | 0.8 | 0.95 |
| W/O | 83.69 | 82.63 | 81.71 | 79.95 | 76.25 | 65.79 |
| SF | 81.77 | 80.65 | 80.04 | 77.65 | 72.66 | 59.56 |
| SA | 81.33 | 81.17 | 78.80 | 77.04 | 72.58 | 58.36 |
| Noise | 81.51 | 80.87 | 79.80 | 77.55 | 73.55 | 59.86 |
| LF | 68.59 | 67.32 | 63.22 | 61.77 | 56.59 | 42.54 |
| Combi | 75.11 | 73.94 | 71.83 | 70.20 | 64.78 | 49.66 |
| Adapt | 78.84 | 78.40 | 76.65 | 75.20 | 71.95 | 61.31 |

Table 4: Overhead measurements for protocol functions across client counts, showcasing the CNN architecture over CIFAR-10 and the MLP over MNIST.

| Measurement | Time Overhead (seconds) on CIFAR-10 | | | | | | Time Overhead (seconds) on MNIST | | | | | |
|---|---|---|---|---|---|---|---|---|---|---|---|---|
| # Clients | 3 | 5 | 8 | 10 | 20 | 30 | 3 | 5 | 8 | 10 | 20 | 30 |
| Cosine Similarity | 0.09 | 0.20 | 0.50 | 0.73 | 2.91 | 5.63 | 0.035 | 0.089 | 0.22 | 0.36 | 1.33 | 3.20 |
| Secure Comparison | 0.013 | 0.039 | 0.089 | 0.147 | 0.56 | 1.11 | 0.013 | 0.035 | 0.096 | 0.14 | 0.55 | 1.06 |
| L2 Normalization | 0.2 | 0.41 | 0.91 | 1.28 | 4.32 | 9.13 | 0.15 | 0.33 | 0.69 | 0.99 | 2.28 | 7.34 |
| Euclidean Distance | 0.049 | 0.13 | 0.28 | 0.51 | 1.73 | 2.55 | 0.022 | 0.058 | 0.12 | 0.42 | 0.78 | 1.89 |

## 4.4 EFFICIENCY OF **SECUREDL**

The comprehensive evaluation of **SecureDL**'s computational efficiency is detailed in Table 4. This table reports the time overhead for critical protocol functions, Cosine Similarity, Secure Comparison, L2 normalization, and Euclidean distance, across varying numbers of clients (3 to 30) on the CIFAR-10 and MNIST datasets. Specifically, it reports measurements for a CNN architecture on CIFAR-10 and an MLP architecture on MNIST. The SVHN and Fashion-MNIST datasets are excluded from this analysis, as they utilize the same model architectures evaluated in this study. A detailed discussion of the performance results and complexity analysis of **SecureDL** is provided in Appendix F.2.

## 5 CONCLUSION

This paper introduces **SecureDL**, the first framework to simultaneously address privacy and security challenges in decentralized machine learning. **SecureDL** is designed to defend against Byzantine attacks even under a dishonest majority by integrating additive secret sharing within a secure multi-party computation framework. Unlike traditional methods based on direct client monitoring, our proposed protocol shifts to a collaborative, privacy-preserving approach that maintains system integrity even when many clients are compromised. Extensive empirical evaluations across diverse datasets demonstrate that **SecureDL** is robust against a wide range of attacks while maintaining high model accuracy. Moreover, performance analysis highlights that the privacy-preserving operations introduce limited overhead, showcasing the efficiency and practicality of the proposed design. Overall, the results indicate that **SecureDL** provides practical privacy and Byzantine robustness for decentralized learning with quantified overheads.

## 6 REPRODUCIBILITY STATEMENT

We provide all the details of our proposed method **SecureDL**, as well as the experimental setup for reproducibility of our results. Additionally, we will release the source code of **SecureDL** framework (upon acceptance) to stimulate further research on privacy-preserving and Byzantine-resilient decentralized learning.

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

## APPENDIX A   MORE DETAILS ON RELATED WORK

This appendix offers detailed explorations of several key algorithms designed to enhance Byzantine resilience in federated and distributed learning environments.

Various strategies have been proposed to defend against Byzantine attacks in FL environments where a central server coordinates training (Blanchard et al., 2017; Chen et al., 2017; Guerraoui et al., 2018; Xie et al., 2019). These Byzantine-resilient optimization algorithms typically rely on robust aggregation rules to combine model updates from clients, mitigating the influence of malicious participants.

Notable approaches include *distance-based techniques* (Pillutla et al., 2022; Elkordy and Avestimehr, 2022), which detect outlier updates based on their deviation from the majority, and *coordinate-wise defenses*, which analyze each dimension of the update independently to filter outliers (Yin et al., 2018). However, both approaches remain vulnerable to subtle, well-crafted attacks that mimic benign behavior (Baruch et al., 2019). Additional strategies, such as those in (Regatti et al., 2020; Cao et al., 2020; Xie et al., 2020b), rely on performance-based criteria using auxiliary validation data.

Distance-based methods like Krum and Multi-Krum (El-Mhamdi et al., 2021) aim to identify a subset of client updates that are closest to each other, discarding outliers before averaging. Coordinate-wise methods, including trimmed mean and median, operate independently on each parameter dimension to limit the influence of adversaries (Yin et al., 2018). While these techniques are simple and scalable, they can be bypassed by adversaries who craft updates that lie within acceptable statistical bounds (Baruch et al., 2019). Performance-based methods, in contrast, evaluate the quality of updates based on local or global validation performance, which can offer stronger resilience but often raise concerns about scalability and data privacy (Cao et al., 2020).

While Byzantine robustness has been actively explored in FL, its application to DL—where no central server exists—has also received growing attention. Early efforts such as ByRDiE (Yang and Bajwa, 2019) and BRIDGE (Fang et al., 2019) extended FL principles to DL settings by employing techniques like trimmed-mean or median aggregation and coordinate-wise stochastic gradient descent to enhance fault tolerance (Baruch et al., 2019). Nonetheless, these systems continue to face challenges when confronted with advanced adversarial behaviors (Guo et al., 2021).

Subsequent methods were developed to improve robustness specifically in decentralized settings. UBAR (Guo et al., 2021), an evolution of the earlier MOZI algorithm (Guo et al., 2020), introduced a two-stage Byzantine-resilient protocol for decentralized learning. The first stage performs Euclidean distance-based neighbor selection, while the second stage evaluates candidate updates using local training data to filter out malicious behavior. By integrating distance- and performance-based strategies and leveraging local datasets at each client, UBAR offers robust protection against a wide range of Byzantine threats, including those outlined in (Guerraoui et al., 2018).

Basil (Elkordy et al., 2022) further improved scalability and efficiency by proposing a performance-based algorithm tailored for decentralized environments. It employs sequential peer selection, memory-assisted filtering and a logical ring topology to identify and isolate adversarial clients. While algorithms like UBAR and Basil demonstrate improved robustness and efficiency, their reliance on access to client updates raises privacy concerns.

Most recently, RAGD (Resilient Averaging Gradient Descent) (Mao et al., 2024) was introduced as a fully decentralized optimization method that is provably robust to local data poisoning attacks. It filters suspicious gradients by comparing each neighbor's update to a local consensus and aggregates them using the Robust Weighted Sum Estimator (RWSE) as its core rule. RAGD offers formal convergence guarantees under convexity assumptions and does not require auxiliary validation datasets. Complementarily, BALANCE (Fang et al., 2024) introduces a local similarity-based filtering mechanism, where each client compares its model with its neighbors' intermediate updates. Updates that deviate beyond a decaying norm-based threshold are excluded, providing robustness against both heterogeneity-induced divergence and targeted poisoning attacks.

Ye et al. (2024) investigate the trade-off between differential privacy and Byzantine robustness in decentralized learning. They focus on coordinate-wise trimmed mean as the core aggregation rule and study its robustness when Gaussian noise is injected into local updates to preserve privacy. The authors derive theoretical conditions under which trimmed mean remains resilient to Byzantine attacks

despite the presence of noise, offering insights into the interplay between privacy and robustness. The work provides heuristic privacy guarantees based on Renyi differential privacy.

Finally, Franzese et al. (2023) propose a committee-based peer-to-peer learning protocol that combines verifiable secret sharing and MPC with existing robust aggregation rules such as RSA, Centered Clipping, and FLTrust. While their framework provides active security against malicious committee members and robustness to client poisoning, it fundamentally relies on an *honest majority* within the aggregation committee, and this assumption cannot be removed by swapping the MPC backend. Since the work does not introduce a new aggregation rule but instead reuses existing ones inside a cryptographic wrapper, we did not include a comparison with SecureDL.

## A.1 Technical Details on Implemented aggregation rules

This appendix provides technical implementation details for the robust aggregation rules used in our experiments for comparison with **SecureDL**. Each method incorporates different strategies to mitigate the influence of Byzantine clients in decentralized learning. For completeness and reproducibility, we outline the core logic and assumptions behind each rule, including how they process received model updates, handle adversarial behavior, and determine their aggregation output.

**BRIDGE:** The implementation of Trimmed-Mean in BRIDGE involves excluding extreme values from the received model updates, allowing the remaining local model updates to contribute to the final average. Additionally, the network should be structured to withstand up to $b$ Byzantine clients. Using a trim parameter $k < \frac{n}{2}$, the server discards the highest and lowest $k$ values, then computes the mean of the remaining $n - 2k$ values for the global model update. To be effective against malicious clients, $k$ must be at least equal to their number, allowing Trim-mean to manage up to nearly $50\%$ malicious clients.

Note that DP-SGD (Ye et al., 2024) adopts the same coordinate-wise trimmed mean aggregation mechanism as BRIDGE. Therefore, we only implement BRIDGE in our evaluation. Moreover, since DP-SGD introduces additional noise for privacy, it typically results in slightly lower accuracy compared to BRIDGE, which operates without a privacy constraint.

**UBAR:** This protocol uses a distance-based metric to filter estimates based on Euclidean distance, effectively narrowing down the pool of model updates to those contributors who are most likely benign, thereby enhancing the reliability of the collective input. This is followed by a performance-based stage, where the protocol evaluates the loss of these estimates, refining the aggregation to include only those with superior performance.

**Krum:** In this approach, each client $i$ is assigned a score $s_i$, calculated by summing the squared Euclidean distances between its update $\boldsymbol{w}_i$ and the updates from other clients. The scoring formula is $s_i = \sum_{\boldsymbol{w}_j} \|\boldsymbol{w}_j - \boldsymbol{w}_i\|_2^2$, where $\boldsymbol{w}_j$ are the updates from a selected set of $n - f - 2$ clients. Here, $n$ represents the total number of clients in the network, and $f$ is the number of Byzantine clients, which are potentially malicious clients. This selection of $n - f - 2$ clients is crucial, as it involves choosing the updates that are closest to $\boldsymbol{w}_i$ in Euclidean distance, effectively filtering out potential outliers or malicious updates. Krum then selects the client with the lowest score to provide the global model update, ensuring the model's resilience against adversarial actions in a scenario with up to $f$ malicious clients.

**Median:** The Median aggregation technique, as described in Yin et al. (2018), chooses the median value of parameters as the consolidated global model, treating each parameter independently. For every $j$-th model parameter, the server ranks the corresponding parameters from $m$ local models, where $m$ is the number of participating models. The $j$-th parameter of the global model is then determined by the median of these ranked values:

$$\text{Global } w_j = \text{Median}(w_{1j}, w_{2j}, \ldots, w_{mj}).$$

The median method can tolerate up to $\frac{n}{2} - 1$ Byzantine clients, where $n$ is the total number of clients in the network. This resilience stems from the fact that as long as the majority of the values are from honest clients, the median will be derived from these honest values, ensuring the integrity of the global model despite the presence of the Byzantine clients.

**BALANCE:** The BALANCE aggregation rule enables each client to filter out suspicious model updates from its neighbors before aggregation. At each communication round $t$, client $i$ receives

model updates $\mathbf{w}_j^{t+1/2}$ from its neighbors $j$. Client $i$ compares each neighbor's update to its own intermediate model $\mathbf{w}_i^{t+1/2}$ using the following rule:

$$\|\mathbf{w}_i^{t+1/2} - \mathbf{w}_j^{t+1/2}\| \le \gamma \|\mathbf{w}_i^{t+1/2}\|$$

where $\gamma$ is a hyperparameter that controls how far an accepted update can be from the client's own model (typically $\gamma < 1$). Only the neighbor updates that satisfy this condition are included in the aggregation. Finally, client $i$ averages its own and the accepted neighbor updates to produce its new model for the next round. Here, $\mathbf{w}_i^{t+1/2}$ is client $i$'s intermediate model, $\mathbf{w}_j^{t+1/2}$ is the received model from neighbor $j$, and $\| \cdot \|$ denotes the Euclidean norm. BALANCE can tolerate up to a constant fraction of Byzantine clients in the neighborhood, as long as the majority of received updates are from honest clients.

**RAGD:** The Resilient Averaging Gradient Descent (RAGD) algorithm enhances robustness by combining local consensus and outlier filtering in the gradient aggregation step. After local communication, each node receives gradients $\mathbf{g}_j$ from its neighbors. Instead of simply averaging these, each node computes a robust aggregated gradient $\hat{\mu}$ by solving:

$$\hat{\mu} = \operatorname{argmin}_\mu \max_j \|\mathbf{g}_j - \mu\|$$

where $\mathbf{g}_j$ is the gradient from neighbor $j$, $\mu$ is the aggregated value being optimized, and $\| \cdot \|$ is the Euclidean distance. This procedure finds a point (the center) that is as close as possible to the farthest gradient, thus reducing the impact of any outlier or poisoned gradient. The node then updates its local model using $\hat{\mu}$ as the gradient in its descent step. RAGD can tolerate up to a fraction $\alpha < 1/2$ of Byzantine clients in each node's neighborhood, provided the underlying graph is sufficiently well-connected.

## APPENDIX B  PRELIMINARIES ON FUNDAMENTAL OPERATIONS FOR SECRET SHARING

This appendix presents the detailed implementation of core functionalities on secret-shared values, including protocols for arithmetic operations, secure comparison, inversion, square root computation, and preprocessing.

### B.1  PERFORMING ADDITION AND MULTIPLICATION ON SHARES

In this section, we examine addition and multiplication as fundamental operations, following the notational conventions established in Randmets (2017).

Secretly shared values $[\![x]\!]$ and $[\![y]\!]$ can be directly added as $[\![x]\!] + [\![y]\!] = ([\![x]\!]_1 + [\![y]\!]_1, \ldots, [\![x]\!]_n + [\![y]\!]_n)$. However, multiplying secret shared values requires network interaction and distinct methodologies for each security environment.

In a dishonest majority setting, the Beaver triples technique (Beaver, 1991) is used for multiplication operations. This method involves pre-distributing shares ($[\![a]\!]$, $[\![b]\!]$, $[\![c]\!]$) of a multiplication triple $(a, b, c)$, where $a$ and $b$ are generated uniformly randomly, and $c = a \cdot b$. Once the shares of the inputs and the triple are received, each computation party computes $[\![\delta]\!] = [\![x]\!]$ - $[\![a]\!]$ and $[\![\epsilon]\!] = [\![y]\!]$ - $[\![b]\!]$ locally and then reveals $[\![\delta]\!]$ and $[\![\epsilon]\!]$ to other parties. The parties can reconstruct $\delta$ and $\epsilon$ using these shares. Since $a$ and $b$ are generated uniformly random, revealing the shares of $\delta$ and $\epsilon$ does not compromise the security of the protocol. Each party then locally computes:

$$[\![w]\!]_i = [\![c]\!]_i + \epsilon \cdot [\![b]\!]_i + \delta \cdot [\![a]\!]_i + \epsilon \cdot \delta$$

where $[\![w]\!]_i$ is a share of the multiplication result calculated by computation party $i$.

### B.2  SECURE COMPARISON PROTOCOL

In the context of secret sharing, a secure comparison protocol enables the comparison of additive shares without revealing the underlying private values. It determines if one value is greater, lesser, or equal to another, producing a binary result that preserves the confidentiality of the inputs. This

protocol is crucial for privacy-sensitive applications, ensuring that only the outcome of the comparison is disclosed, thereby maintaining the integrity and secrecy of each party's input.

In this work, we have implemented the secure comparison protocol as described by Makri et al. (2021). Their protocol ensures perfect security over the arithmetic ring $\mathbb{Z}_M$ for positive integers in the range $[0, 2^l - 1]$, where $l \in \mathbb{N}$ and $2^{l+1} < M = 2^k$. The protocol's adaptability to different arithmetic models and its effectiveness in settings with a dishonest majority are its key strengths.

The basic idea of the protocol is that the sum of two secret shared values $a, b \in \mathbb{Z}_M$ modulo $M$ is less than $a$ and less than (LT) $b$, iff $a + b$ is reduced by $M$ :

$$(a + b) \bmod M = a + b - M \cdot \text{LT}(a + b \bmod M, a)$$
$$= a + b - M \cdot \text{LT}(a + b \bmod M, b)$$

The protocol's online phase involves minimal rounds and arithmetic operations, while the offline phase requires additional resources such as edaBits (Escudero et al., 2020).

### B.3 INVERSION OPERATION

Our multi-party computation framework does not natively support inversion operations. To address this limitation, we adopt an iterative approximation approach, similar to the method proposed by Nardi et al. (2012). This strategy enables us to perform inversion using fundamental operations such as addition and multiplication, effectively bypassing the need for direct inversion on secret-shared inputs.

The approach centers around finding a value $x$ that serves as the inverse of a given value $X$. To achieve this, we define a function $f(x)$ such that $f(x) = 0$ when $x$ satisfies the inverse relationship for $X$. The function is formulated as follows:

$$f(x) = \frac{1}{X} - x$$

To compute the root of $f(x)$, we leverage the Newton-Raphson method (Agresti, 2003), which provides a stable numerical iterative approximation. The method is represented as:

$$x_{n+1} = 2x_n - x_n \cdot y_n \quad \text{where } x_0 = c^{-1},$$
$$y_{n+1} = 2y_n - y_n^2 \quad \text{and } y_0 = c^{-1} \cdot X,$$

where $c$ is a constant, and $y_n = x_n \cdot X$. After approximately 15 iterations, $x_n$ converges to an accurate approximation of the inverse of $X$ (Ghavamipour et al., 2022).

### B.4 SQUARE ROOT COMPUTATION

In our protocol, square root computation is essential but not directly supported by the multi-party computation framework. To address this limitation, we employ an approximation technique based on the Newton-Raphson method, which enables square root computation using only basic operations such as addition and multiplication (Press, 2007).

The Newton-Raphson method iteratively refines an initial approximation of the square root using a function and its derivative. To compute the square root of a number $Y$, the method uses the function $f(x) = x^2 - Y$, aiming to find an $x$ such that $f(x) = 0$. The iterative update is defined as follows:

$$x_{n+1} = \frac{1}{2} \left( x_n + \frac{Y}{x_n} \right),$$

where $x_n$ is the current approximation and $x_{n+1}$ is the updated guess. Starting with an initial guess $x_0$, the method iteratively applies this update. The term $\frac{Y}{x_n}$ provides an increasingly accurate approximation of the square root of $Y$ when averaged with $x_n$.

After approximately 15 iterations (as implemented in our framework), $x_n$ converges to an accurate approximation of the square root of $Y$.

### B.5    Preprocessing

Our protocol is designed to be compatible with Message Authentication Codes (MACs) and Beaver triples generation, following approaches similar to those in Keller (2020). Our implementation necessitates multiplication triples over $\mathbb{Z}_{2^k}$, where $k$ is a parameter of the protocol. For the secure comparison protocol, when $k = 1$, our protocol incorporates an optimized variant of the two-party TinyOT protocol (Wang et al., 2017). For larger choices of $k$ (i.e., $k = 32$ in this work), we base our costs on the MASCOT protocol (Keller et al., 2016), noting that despite MASCOT's communication complexity being in $O(k^2)$, it still offers the lowest costs for all the table sizes we have considered, particularly for $k = 32$.

In our analysis, we concentrate primarily on the dynamics of the online phase of the protocol. We omit the details of the offline phase, encompassing the generation and distribution of MACs and Beaver triples, which can be found in Cramer et al. (2018).

## Appendix C    Threat model and Problem Statement

In this work, we address scenarios in which a majority of clients may behave dishonestly. Such malicious clients seek to undermine the learning process or introduce bias by sending deceptive or false updates. Furthermore, they may collaborate to amplify the effectiveness of their attacks, including strategies like model poisoning and inference attacks.

In collaborative learning environments, an honest client aims to obtain an aggregated model update from multiple local datasets, rather than a specific model from another client. This access to only the aggregated update provides two key privacy benefits. Firstly, it prevents malicious attempts targeting client models by masking details specific to individual local models, effectively hindering adversaries from identifying and exploiting precise data sources. Secondly, it diminishes the impact of any single client's input by combining contributions from multiple clients within the aggregated model. This reduction in the prominence of individual contributions serves to obscure their visibility, thus decreasing the likelihood of attacks that target unique datasets.

In FL, a central server aggregates the model updates received from the clients, relaying only the aggregated outcome to those clients. Thus, the server is the only entity with access to each model's (non-aggregated) updates. Conversely, in DL, with a shift to a completely decentralized training, the elimination of the central server increases the attack surface. This increase is due to the need to distribute updates across neighboring clients for on-client aggregation, contrasting sharply with FL. Therefore, in DL, each client, with access to collaborator updates can potentially perform privacy attacks against their collaborators such as inference attacks (Pasquini et al., 2023).

Collaborative learning systems are particularly vulnerable to Byzantine client disruptions, where even a single malicious participant can compromise the entire learning process by corrupting shared model updates (Blanchard et al., 2017; Raynal et al., 2023). In FL settings, various research studies have been conducted to enable the server to detect and mitigate Byzantine threats. However, in DL, the absence of centralized control within decentralized architectures presents unique challenges. In a decentralized setting, each client in the network must independently address these challenges, which complicates the detection and mitigation of malicious activities.

In this study, our objective is to develop a DL method that maintains Byzantine robustness against malicious clients without compromising accuracy or efficiency. The Byzantine robustness of our protocol is achieved through a privacy-preserving aggregation rule. This rule prevents clients from accessing other clients' data in plain form; instead, only aggregated values are revealed to them.

## Appendix D    Experimental Setup

This appendix provides detailed information about the experimental setup, including the datasets, model architectures, system environment, and data partitioning strategies used in our evaluations.

To account for randomness and variability, all experiments were conducted at least five times using different random seeds. The reported results represent the average performance across these runs. Given the stochastic nature of decentralized learning and the use of approximation techniques in

**SecureDL**, the outcomes may vary slightly depending on factors such as client ordering, model initialization, and random selection processes. Although we strive for consistency, such fluctuations are inherent to the design of the framework and should be taken into account when interpreting the reported accuracy values.

## D.1 DATASETS

We evaluated our protocol using four popular deep learning datasets: MNIST (LeCun, 1998), Fashion-MNIST (Xiao et al., 2017), SVHN (Netzer et al., 2011), and CIFAR-10 (Krizhevsky et al., 2009) with independent and identically distributed partitioning. We randomly split the training set into subsets of a desired size (mostly evenly) and allocate each subset to each client. The MNIST dataset contains 60,000 training and 10,000 testing grayscale images of handwritten digits, each with a resolution of $28 \times 28$ pixels. It is evenly distributed across ten classes, with each class represented by 6,000 images in the training set and 1,000 in the test set. Fashion-MNIST is a 10-class fashion image classification task, which has a predefined training set of 60,000 fashion images and a testing set of 10,000 fashion images. The SVHN dataset, sourced from Google Street View house numbers, includes 99,289 color images across ten classes. It comprises 73,257 training and 26,032 testing images, all standardized to a resolution of $28 \times 28$ pixels. CIFAR-10, a collection of color images, offers a diverse challenge with its 50,000 training and 10,000 testing examples spread across ten distinct classes. Each example in this dataset is a color image representing one of these classes.

## D.2 MODEL ARCHITECTURES

We employed a Convolutional Neural Network (CNN) for the CIFAR-10 and SVHN datasets, and a Multi-Layer Perceptron (MLP) for the Fashion MNIST and MNIST datasets. As detailed in Table 5, the CNN receives a $3 \times 32 \times 32$ input and processes it through a sequence of convolutional layers with Group Normalization (GN) and ReLU activations, interleaved with max pooling and dropout layers, with dropout rates increasing from 0.2 to 0.5. The convolutional blocks progressively expand the number of channels from 32 to 128, followed by two fully connected layers—the first equipped with GroupNorm, ReLU, and dropout, and the second producing the final 10-class output. This architecture comprises approximately 1.4 million parameters and is trained with a learning rate of 0.002 and a batch size of 128. We selected this design to ensure a fair comparison with state-of-the-art decentralized and federated learning methods that adopt similar CNN backbones.

For the simpler grayscale datasets, Fashion MNIST and MNIST, we employed a two-layer MLP with sigmoid activations. This model consists of an input layer (flattened to 784 units), followed by hidden layers with 300 and 100 units, and a final 10-unit output layer. It is optimized with a learning rate of 0.01 and the same batch size. With around 110,000 parameters, the MLP provides a lightweight yet effective solution for lower-resolution image classification tasks.

It is worth emphasizing that while these specific architectures are chosen for consistency with prior work and fairness in evaluation, the SecureDL framework itself is not restricted to small or medium-scale models. In particular, the protocol is fully compatible with deep architectures containing tens of millions of parameters, such as ResNet or DenseNet, making it applicable to large-scale tasks as well.

## D.3 LIMITATIONS

While **SecureDL** demonstrates strong privacy and Byzantine robustness in our experimental evaluation, there remain aspects that can be further strengthened. These aspects highlight the assumptions underlying our current design and point to promising directions for future work to enhance scalability, adaptability, and applicability in real-world deployments. We outline the most salient considerations below.

### D.3.1 MULTI-THREADING ASSUMPTIONS AND REAL-WORLD DEPLOYMENT

The implementation of **SecureDL** relies on a multi-threading environment where each thread represents a client in the decentralized learning setting. This architectural choice assumes that client operations are independent and efficiently managed by the threading model. Many approaches, including ours, are evaluated under controlled experimental settings rather than real-world deployments. Factors such as client drop-out, network unreliability, asynchronous updates, and system failures can significantly impact overall system performance and scalability. Consequently, the real-world

| Layer Type | Size / Type |
|---|---|
| Input | $3 \times 32 \times 32$ |
| Conv + GroupNorm + ReLU | $3 \times 3 \times 32$ / GN(32) |
| Conv + GroupNorm + ReLU + Max Pool + Dropout(0.2) | $3 \times 3 \times 32$ / GN(32) |
| Conv + GroupNorm + ReLU | $3 \times 3 \times 64$ / GN(64) |
| Conv + GroupNorm + ReLU + Max Pool + Dropout(0.3) | $3 \times 3 \times 64$ / GN(64) |
| Conv + GroupNorm + ReLU | $3 \times 3 \times 128$ / GN(128) |
| Conv + GroupNorm + ReLU + Max Pool + Dropout(0.4) | $3 \times 3 \times 128$ / GN(128) |
| Fully Connected + GroupNorm + ReLU + Dropout(0.5) | 128 / GN(128) |
| Fully Connected | 10 |

| Layer Type | Size / Type |
|---|---|
| Input | $1 \times 28 \times 28$ (flattened to 784) |
| Fully Connected + Sigmoid | 300 |
| Fully Connected + Sigmoid | 100 |
| Fully Connected | 10 |

Table 5: Model architectures used in our experiments. The top table shows the CNN for CIFAR-10 and SVHN; the bottom table shows the MLP for Fashion MNIST and MNIST.

performance of **SecureDL** may vary when these multi-threading assumptions do not hold, and further investigation is needed to assess its robustness in more heterogeneous or distributed environments.

### D.3.2  OFFLINE PHASE ASSUMPTIONS

Similar to other works in the secret sharing and MPC literature, our approach assumes that the offline phase of secret sharing protocols (such as Beaver triple generation) is completed prior to the actual execution of the decentralized learning framework. As a result, our computational performance measurements consider only the online phase. This assumption may underestimate the practical costs and complexities associated with the offline phase, which could affect overall system efficiency and latency in real-world deployments.

### D.3.3  DROPOUT CONSIDERATIONS

Additive secret sharing, as adopted in **SecureDL**, requires the availability of all shares for correct reconstruction and thus does not natively tolerate missing participants. Nevertheless, this does not imply that the protocol becomes inoperable under client churn. If dropout occurs prior to the start of a round, the absent client's shares can be excluded and training proceeds with the active set. If dropout arises during an ongoing round, the secure computation cannot be completed; however, the round may be terminated and the participants can reestablish shares among the remaining clients before resuming training. These considerations indicate that dropout can be accommodated within the **SecureDL** framework through standard resharing techniques, ensuring that dynamic participation is addressed without undermining correctness or privacy.

### D.3.4  PARAMETER SENSITIVITY (COSINE THRESHOLD AND TRUST NORMALIZATION)

The choice of the cosine similarity threshold is highly dependent on both the model architecture and the complexity of the dataset. Importantly, the threshold value itself is private, since revealing it could give adversaries an advantage in adaptive attacks. In our study, we employ secure comparison to evaluate updates against the threshold and explore several strategies for threshold selection, including fixed thresholds, adaptive tuning, and dynamic adjustment based on global statistics. Each strategy has theoretical appeal but also practical limitations. For instance, Cao et al. (2020) employ a non-negative cosine similarity rejection threshold (i.e., accepting updates with non-negative similarity), but overly permissive thresholds (e.g., 0.5) may assign low trust scores to benign updates, reducing their contribution to the aggregation despite acceptance. Conversely, fully adaptive thresholds often introduce instability or require complex coordination, which can be impractical in decentralized settings.

Through empirical evaluation, we found that a lightweight trial-and-error approach, performed locally by each client using a small validation subset, provides a more reliable and computationally efficient solution. Our results indicate that **SecureDL** remains robust across a reasonable threshold range (e.g., 0.8–0.95 for most datasets considered), with expected trade-offs: lower thresholds increase the likelihood of admitting adversarial updates, whereas overly strict thresholds risk excluding benign ones. Finally, the softmax-based trust score normalization further stabilizes aggregation performance across diverse datasets and attack types.

## APPENDIX E   DETAILED DESCRIPTION OF ATTACKS

This appendix provides detailed descriptions of the poisoning and model manipulation attacks used in our evaluations.

**Sign-Flipping Attack (SF).**   In this attack, the adversary reverses the direction of the local update vector by multiplying it by $-1$. Formally, if the original update is $\hat{w}_i^k$, the transmitted update becomes $w_i^k = -\hat{w}_i^k$. This inversion drives the global model in the wrong direction and can significantly delay or degrade convergence.

**Gaussian Attack (Noise).**   Here, a Byzantine client replaces its update with a random vector sampled from a Gaussian distribution with mean and variance 0.1, that is, $w_i^k \sim \mathcal{N}(0.1, 0.1)$. The goal is to inject noise into the aggregation process, weakening the model's signal-to-noise ratio.

**Scaling Attack (SA).**   In the scaling attack (Bagdasaryan et al., 2020), the adversary amplifies its model update by a large scalar. This inflated update, when included in the aggregation, can disproportionately influence the global model. It is particularly effective when the aggregation method is sensitive to update magnitudes.

**Label-Flipping Attack (LF).**   Label flipping is a form of data poisoning where an attacker inverts the labels of local training samples. For a classification task with $L$ classes and a true label $l$, the flipped label becomes $L - 1 - l$. This technique aims to mislead the model during training without changing update magnitudes or directions.

**Combination Attack (Combi).**   To evaluate resilience against stronger adversaries, we introduce a combination attack that fuses multiple strategies—such as sign-flipping, noise injection, scaling, and label flipping—into a single client. This creates diverse and unpredictable malicious behavior that stresses the defense mechanisms more comprehensively.

**Adaptive Attack (AA).**   Adaptive attacks craft malicious updates by optimizing against the known aggregation rule (Fang et al., 2020). We implement a dual-objective attack where a malicious client perturbs its local update $g_i$ to generate $g_{\text{adv}}$ by solving:

$$\min_{g_{\text{adv}}} -\|g_{\text{adv}} - g_i\|^2 + \lambda_{\text{cos}} \cdot \max(0, \theta - \cos(g_{\text{adv}}, g)) + \lambda_{\text{dist}} \cdot \|g_{\text{adv}} - g\|^2,$$

where $g$ is the trusted global model, $\theta$ is the minimum cosine similarity threshold, and $\lambda_{\text{cos}}, \lambda_{\text{dist}}$ are dynamically adjusted penalties. This formulation maximizes deviation from benign behavior while maintaining sufficient cosine similarity and controlled Euclidean distance to evade defense filters. The penalties are adaptively tuned during optimization to meet dynamic constraints without weakening the attack's strength.

**SecureDL** robustness against a broad range of Byzantine behaviors fundamentally stems from the privacy-preserving layer integrated into the decentralized training framework. This layer employs secret sharing to prevent exposure of individual model updates. As a result, no client, including malicious ones, can access the raw updates of others; only the final aggregated result is revealed. This architectural constraint significantly limits attackers' ability to perform targeted or coordinated manipulations based on inspecting benign updates. Consequently, several well-known Byzantine attacks, such as "a little is enough" (Baruch et al., 2019), Krum, and Trimmed Mean attacks (Fang et al., 2020), are inherently prevented by our defense mechanisms, as these attacks fundamentally rely on access to other clients' updates—an attack surface our protocol eliminates by design. Moreover,

other attacks, such as the inner-product manipulation (IPM) attack (Xie et al., 2020a), are naturally covered as simple instances within our broader adaptive attack model, which is more general and sophisticated in scope.

## APPENDIX F    MORE DETAILS ON EVALUATION

This appendix provides additional experimental results and analysis to further evaluate the performance, robustness, and scalability of **SecureDL** under more challenging and realistic conditions. We begin by examining the impact of non-iid data distributions on the resilience of **SecureDL** against Byzantine attacks. This is followed by a detailed analysis of the computational overhead introduced by key protocol components, including similarity computations, secure comparisons, and normalization steps. Finally, we present a complexity analysis of **SecureDL** in terms of both computation and communication, highlighting how its security mechanisms scale with the number of participating clients.

### F.1    SECUREDL ROBUSTNESS AGAINST NON-IID DATA DISTRIBUTIONS

To assess the robustness of **SecureDL** in decentralized learning environments, we evaluated its performance across varying degrees of non-iid data distributions, with a fixed setting of 4 Byzantine attackers among 10 total clients. The experiments were conducted on four datasets: MNIST, VHDS, Fashion-MNIST, and CIFAR-10, each subjected to controlled increases in data skewness. Table 3 presents the accuracy of **SecureDL** under different attack types and skewness levels.

Across all datasets, **SecureDL** demonstrated strong resilience to increasing non-iid distributions. On the MNIST dataset, the model's accuracy without attacks declined modestly from 97.10% in the iid setting to 80.32% at extreme skewness (0.95), while maintaining 77.85% even under the challenging Noise attack. A similar trend was observed for the VHDS dataset, where the accuracy under no attack decreased from 94.64% to 89.55%, and remained above 86% even under targeted attacks such as LF and SA. These results confirm that **SecureDL** effectively sustains high performance across datasets with structured information, despite growing data heterogeneity. In contrast, the Fashion-MNIST dataset showed remarkable stability, with performance across all skewness levels fluctuating by less than 0.2%, including under attack scenarios like Noise and Adapt. This indicates that for simpler datasets, **SecureDL** successfully neutralizes the effects of data distribution shifts and adversarial behavior with minimal performance degradation.

Although CIFAR-10 posed a greater challenge due to its higher complexity, **SecureDL** still preserved reasonable robustness. While the accuracy without attacks decreased from 83.69% under iid conditions to 65.79% at high skewness, the model consistently outperformed expectations compared to standard baselines in non-iid federated settings. Under severe attacks such as Combi and Noise at 0.95 skewness, **SecureDL** achieved 49.66% and 59.86% accuracy respectively, reflecting substantial resilience. Notably, the lowest accuracy was observed under the LF attack on CIFAR-10 with high skewness, where the accuracy dropped to 42.54%. This outcome underscores the difficulty of defending against semantic-targeted attacks in visually complex datasets, especially under extreme data heterogeneity. However, even in this setting, **SecureDL** outperformed baseline methods, and the controlled accuracy drop suggests that its aggregation rule still manages to dampen the compounded impact of adversarial updates and skewed distributions.

Overall, the consistent performance trends across these datasets highlight that **SecureDL** is capable of handling diverse challenges associated with real-world decentralized learning. These findings confirm that **SecureDL** achieves strong robustness to non-iid data distributions while effectively defending against Byzantine threats, making it a promising solution for practical federated learning deployments where data heterogeneity and adversarial conditions are common.

### F.2    PERFORMANCE AND COMPLEXITY ANALYSIS OF **SECUREDL**

The data presented in Table 4 reveals a consistent upward trend in computational overhead as the number of clients increases. For the CIFAR-10 dataset, the execution time for Cosine Similarity rises from 0.09 seconds with three clients to 5.63 seconds with thirty clients, representing an increase of over 60 times. Euclidean Distance also shows a clear increase, growing from 0.049 seconds to 2.55 seconds. Although Secure Comparison increases from 0.013 seconds to 1.11 seconds, it

remains relatively lightweight compared to similarity and distance measures. L2 normalization exhibits a noticeable rise as well, expanding from 0.2 seconds to 9.13 seconds as the number of clients increases. These results highlight that similarity and distance computations—particularly Cosine Similarity—become dominant contributors to processing overhead as the network scales, posing challenges for the scalability of decentralized systems such as **SecureDL**.

A similar pattern is observed in the MNIST dataset, although with consistently lower absolute overheads due to its simpler model architecture. Cosine Similarity increases from 0.035 seconds with three clients to 3.20 seconds with thirty. Euclidean Distance rises from 0.022 seconds to 1.89 seconds, following the same upward trend. Secure Comparison grows more modestly, from 0.013 seconds to 1.06 seconds, while L2 normalization increases from 0.15 seconds to 7.34 seconds. These variations reflect differences in model complexity: evaluations on MNIST employed a two-layer Multi-Layer Perceptron, whereas CIFAR-10 experiments utilized a deeper Convolutional Neural Network, as detailed in Table 5. This distinction clarifies the relative scaling behaviors and demonstrates **SecureDL**'s adaptability across different model and data environments.

Overall, the overhead analysis confirms that computations involving similarity and distance measures can significantly affect scalability as the number of clients increases. Nevertheless, **SecureDL** maintains a balance between security guarantees and computational efficiency, supporting its practical viability for secure and scalable decentralized machine learning applications across diverse settings.

## F.3 COMPUTATIONAL AND COMMUNICATION COMPLEXITY

In the **SecureDL** protocol within a multi-party computation framework, the computational complexity predominantly arises from several key operations involving all participating clients. Initially, each of the $n$ clients engages in additive secret sharing, computing and distributing secret shares of their inputs to the others, which results in $n \times (n-1)$ computational and communication tasks, leading to a complexity of $O(n^2)$. Subsequently, the clients compute cosine similarities, where each client calculates the similarities between its own model update and every other client's update, necessitating $n^2$ computations. Despite identical calculations being required between client pairs, the security requirements demand separate computations, maintaining an overall complexity of $O(n^2)$. The protocol also includes secure comparisons of these cosine similarities against a threshold, preserving the $O(n^2)$ scaling. Following this, accepted model updates are normalized using secure L2 normalization, a process that, in the worst case, involves up to $n^2$ secure multiplications and divisions per round. For each accepted and normalized update, the protocol further computes the Euclidean distance to the receiver's model using secure subtraction, squaring, summation, and square root operations, again scaling as $O(n^2)$. The assignment and normalization of trust scores, as well as the weighted aggregation of updates based on these scores, are also carried out for every client pair, preserving the quadratic scaling. Consequently, the cumulative computational complexity of the **SecureDL** protocol is determined to be $O(n^2)$, reflecting the quadratic scaling in computational demands with the increase in the number of participating clients.

The overall communication complexity is influenced by multiple computational steps, each with its own impact on the total communication requirements. Initially, clients engage in secret sharing with linear complexity $O(n)$, where $n$ is the number of clients. This is followed by the cosine similarity computation and L2 normalization, each involving $k$ Beaver triple multiplications per pairwise computation among the $n$ clients, resulting in a significant communication load of $O(kn^3)$ for each step. The computation of Euclidean distances and the subsequent trust score assignments, including normalization and weighted aggregation, similarly require secure multiplications and divisions for each client pair and thus contribute to the $O(kn^3)$ complexity as well. Additionally, the protocol employs a secure comparison operation, which, based on the Rabbit protocol's complexity of $\mathcal{O}(\ell \log \ell)$ and being performed $n^2$ times (once for each pair of clients), contributes significantly to the communication complexity. Assuming that the bit length $\ell$ of the integers involved is constant and not dependent on $n$, this step adds $O(n^2 \ell \log \ell)$ to the overall complexity. Therefore, when combining these complexities, the overall communication complexity of the **SecureDL** protocol is dominated by the cosine similarity, L2 normalization, Euclidean distance computation, trust score normalization, and weighted aggregation steps, culminating in a total complexity of approximately $O(kn^3)$. This underlines the protocol's considerable communication demands, scaling cubically with the number of clients due to the Beaver triple operations required for secure multiplications and the overall pairwise secure computations needed to preserve both privacy and robustness throughout the aggregation process.

## APPENDIX G    CONVERGENCE PROOF DETAILS

In this section, we provide the full convergence proof for **SecureDL**, adapted and extended based on Cao et al. (2020).

The convergence analysis relies on the following assumptions:

**Assumption 1.** *The expected loss function $f(D, \mathbf{w})$ is $\mu$-strongly convex and differentiable over $\mathbb{R}^d$, with an $L$-Lipschitz continuous gradient. Specifically, for all $\mathbf{w}, \widehat{\mathbf{w}} \in \mathbb{R}^d$:*

$$f(D, \widehat{\mathbf{w}}) \geq f(D, \mathbf{w}) + \langle \nabla f(D, \mathbf{w}), \widehat{\mathbf{w}} - \mathbf{w} \rangle + \frac{\mu}{2}\|\widehat{\mathbf{w}} - \mathbf{w}\|^2,$$

$$\|\nabla f(D, \widehat{\mathbf{w}}) - \nabla f(D, \mathbf{w})\| \leq L\|\widehat{\mathbf{w}} - \mathbf{w}\|,$$

*where $\nabla$ denotes the gradient, $\|\cdot\|$ the $L_2$-norm, and $\langle \cdot, \cdot \rangle$ the inner product.*

**Assumption 2.** *The empirical loss function $f(D, \mathbf{w})$ is $L_1$-Lipschitz probabilistically, and the gradient $\nabla f(D, \mathbf{w})$ is bounded.*

**Assumption 3.** *Each client's local dataset $D_i$ is independently drawn from a common distribution $\xi$.*

**Assumption 4.** *Errors introduced by secure computation protocols (e.g., secure comparison, secure aggregation) are negligible compared to statistical noise inherent to decentralized stochastic learning.*

**Proof of Theorem 1:**

Under the above assumptions, we can adapt the convergence proof from Cao et al. (2020) to **SecureDL**. In FLtrust, after rejecting malicious updates based on cosine similarity, the surviving updates are normalized and scaled by their positive cosine similarities before aggregation. In **SecureDL**, the main differences are:

- Updates are accepted based on passing a secure cosine similarity threshold test ($\tau$). - Accepted updates are L2-normalized to match the receiver's model magnitude. - A trust score based on secure Euclidean distance is computed for each accepted update. - Aggregation is performed using these normalized trust scores.

Formally, in **SecureDL**, the update aggregation weight for client $i$ is:

$$\varphi_i = \frac{Step(\cos(\mathbf{w}_i, \mathbf{w}_0), \tau)}{|\mathcal{S}|} \quad \text{(binary acceptance via thresholding)}$$

followed by scaling based on trust scores:

$$\varphi_i' = \frac{s_i}{\sum_{j \in \mathcal{S}} s_j},$$

where $s_i$ is the trust score assigned to $i$ based on the inverse Euclidean distance.

This modification satisfies the required conditions for convergence:

1. $\sum_{i \in \mathcal{S}} \varphi_i' = 1$, 2. $0 < \varphi_i' < 1$ for all accepted updates, 3. Larger trust scores are given to updates closer (in distance) to the receiver's current model, reducing adversarial influence.

Given these properties, the proofs of Lemmas 1, 2, and 3 in Cao et al. (2020) still apply. Specifically:

- Lemma 1 bounds the deviation of the aggregated update from the benign average.
- Lemma 2 ensures that the trusted updates still approximate the true gradient.
- Lemma 3 connects bounded deviation to convergence under strong convexity and smoothness.

Thus, we conclude that **SecureDL** achieves convergence with the following guarantee:

$$\|\mathbf{w}^t - \mathbf{w}^*\| \leq (1 - \rho)^t \|\mathbf{w}^0 - \mathbf{w}^*\| + \frac{12\alpha\Delta_1}{\rho},$$

where $\mathbf{w}^t$ is the global model at round $t$, $\mathbf{w}^*$ is the optimal model, $\alpha$ is the global learning rate, and $\rho$, $\Delta_1$ are constants defined based on the gradient smoothness and deviation bounds.

In particular, when $|1 - \rho| < 1$, as $t \to \infty$:

$$\lim_{t \to \infty} \|\mathbf{w}^t - \mathbf{w}^*\| \leq \frac{12\alpha\Delta_1}{\rho},$$

thus confirming the convergence of **SecureDL** even in the presence of Byzantine clients.

## APPENDIX H    PRIVACY ANALYSIS

In this section, we provide the formal simulation-based proof for the privacy guarantees stated in Theorem 2.

The **SecureDL** protocol achieves privacy against malicious adversaries through a combination of secure secret sharing and information-theoretically secure message authentication codes (MACs). These MACs ensure integrity and authentication without leaking any information about the protected messages (Bellare et al., 1996). Their information-theoretic security guarantees robustness even against computationally unbounded adversaries.

Our aggregation protocol involves several key operations: multiplication, addition, inversion, square root computation, vector norm computation, cosine similarity assessment, and L2 normalization. For each operation, we provide an individual simulation-based proof. The inherent randomness of the secret sharing mechanism ensures that the output shares from each operation remain computationally indistinguishable from random values, even when reused as inputs for subsequent steps.

This property allows us to securely compose all operations without leakage. Consequently, there exists a global simulator that can simulate the adversary's view throughout the entire protocol execution. Therefore, the **SecureDL** protocol maintains the privacy of honest clients' inputs even in the presence of malicious adversaries controlling up to $n - 1$ participants.

*Proof of Theorem 2.* Security against malicious adversaries is achieved through the incorporation of unconditionally secure, information-theoretically sound MACs (Cramer et al., 2018; Damgård et al., 2012). These MACs ensure that each piece of data in the computation is authenticated, preventing unauthorized modifications by verifying the integrity and authenticity of the data throughout the process. Inspired by these principles, the **SecureDL** protocol extends this approach by employing MACs not only as a means of data authentication but also as a critical component in protecting privacy and confidentiality. By simulating the perspective of a semi-honest adversary, we initially establish the security foundation of our system. Subsequently, we enhance our protocol to defend against active adversaries by integrating these robust MACs, ensuring that each operation within the protocol remains secure and trustworthy, thus preserving the integrity of the entire computational process.

The functionality of the protocol $\pi$, denoted as $f(x, y) := w$, is deterministic and ensures correctness. This correctness is verified by computing the sum of all outputs from $n$ parties and evaluating this sum modulo $\mathbb{Z}_{2^k}$, specifically, $(w_1 + w_2 + \ldots + w_n \mod \mathbb{Z}_{2^k})$. The protocol specifies for each party $P_i$ a corresponding component of $f(x, y)$, $f_i(x, y) := w_i$, where $w_i$ is the output computed by party $P_i$.

Define the view of party $P_i$ during the execution of protocol $\pi$ as:

$$\text{view}_i^\pi(x, y) := (x_i, y_i, r_i, m_1, m_2, \ldots, m_t) \in \mathbb{Z}_{2^k},$$

where $r_i$ represents the outcome of the $P_i$ internal coin tosses and $m_1, m_2, \ldots, m_t$ denote the received messages.

To prove that $\pi$ privately computes $f$, we must show that there exists a probabilistic polynomial-time simulator $S$ for every $I$ such that:

$$\{S(x_I, f_I(x, y))\}_{x,y} \stackrel{\text{perf}}{\equiv} \{\text{view}_I^\pi(x, y)\}_{x,y}.$$

The **SecureDL** algorithm involves three key operations on secret-shared data: Cosine similarity computation, comparison over secret shared data and L2normalization.

The actual view of $P_I$, for the computation is as follows:

$$\text{view}_I^\pi(\vec{x}) = (\vec{x}, m_1^{cosine}, m_2^{cosine}, \ldots, m_I^{cosine}, m_1^{comp},$$
$$m_2^{comp}, \ldots, m_n^{comp}, m_1^{l2norm}, m_2^{l2norm}, \ldots,$$
$$m_I^{l2norm})$$

where $\vec{x}$ denotes the vectors of additive shares of the inputs held by each honest parties, $m_i^{cosine}$ are the messages exchanged during the cosine similarity computation, $m_i^{comp}$ are the messages exchanged for the comparison operation and $m_i^{l2norm}$ are the messages exchanged for the L2 normalization computation.

Given the additive sharing scheme, all elements involved in $\text{view}_I^\pi(\vec{x}$ are independently and uniformly distributed within $\mathbb{Z}_{2^k}$, ensuring that from $P_I$'s perspective, all components are indistinguishable from random. This setup, based on the privacy-preserving properties of cosine similarity, secure comparison and L2 normalization operations, allows $S_I$ to simulate a plausible set of interactions that $P_I$ (or any group of honest parties) would observe, without revealing any private information. The simulator $S_I$ thus constructs a simulated view that includes simulated messages for each step of the norm computation, ensuring these simulated components are indistinguishable from those in a real protocol execution.

We construct a simulator, denoted $S_I$, for simulating the collective view of honest parties within the execution of **SecureDL** protocol. The role of $S_I$ is to create a plausible simulation of the protocol execution as it would appear to the honest parties, focusing on the data exchanges and computations that occur without assuming control over the adversary's direct observations.

In the case of cosine similarity computation, $S_I$ simulates the messages exchanged during this step, such as $\tilde{m}_1^{cosine}, \tilde{m}_2^{cosine}, \ldots, \tilde{m}_n^{cosine}$. This simulation is consistent with the privacy assurances specified in Theorem 7, making these simulated messages indistinguishable from random values in $\mathbb{Z}_{2^k}$ to any observer, thereby maintaining the integrity of the privacy guarantees.

During the comparison, $S_I$ constructs messages such as $\tilde{m}_1^{comp}, \tilde{m}_2^{comp}, \ldots, \tilde{m}_n^{comp}$, simulating the inter-party communications. **SecureDL** utilizes the Rabbit secure comparison protocol (Makri et al., 2021), which has perfect security within $\mathbb{Z}_{2^k}$. Therefore, to any observer, these simulated messages are indistinguishable from random values within $\mathbb{Z}_{2^k}$, thereby upholding the integrity of the privacy guarantees.

Finally, during the L2 normalization computation, $S_I$ constructs messages such as $\tilde{m}_1^{l2norm}, \tilde{m}_2^{l2norm}, \ldots, \tilde{m}_n^{l2norm}$, simulating the messages exchanged during. These simulated messages are crafted to be statistically indistinguishable from those in a genuine protocol run, aligning with the privacy-preserving methodology of the square root function described in Theorem 8.

The simulated view $S_I$ provides for the protocol includes the inputs of all such parties, denoted as $x_1, x_2, \ldots, x_m$, where $m$ is the number of honest parties. Thus, the complete simulator view is represented as:

$$S_I(\bar{x}, f_I(\bar{x})) = (\tilde{\bar{x}}, \tilde{m}_1^{cosine}, \tilde{m}_2^{cosine}, \ldots, \tilde{m}_n^{cosine}, \tilde{m}_1^{comp}$$
$$, \tilde{m}_2^{comp}, \ldots, \tilde{m}_n^{comp} \tilde{m}_1^{l2norm}, \tilde{m}_2^{l2norm}, \ldots,$$
$$\tilde{m}_n^{l2norm})$$

Every component of $S_I$ is independent and uniformly random within $\mathbb{Z}_{2^k}$). As $S_I$ is designed to select values that are not only independently and uniformly random but also drawn from the same distribution as those in $\text{view}_I^\pi$, these components become indistinguishable. This indistinguishability, ensures that they resemble random values, thus reinforcing the privacy aspect of the protocol.

$\square$

**Theorem 3** (privacy w.r.t semi-honest behavior). *Consider a secure multi-party computation protocol $\pi$ designed to compute a function $f(x, y)$ utilizing the Beaver multiplication approach (detailed in Section B.1). This protocol involves $n$ parties, denoted as $P_1, P_2, \ldots, P_n$, with a subset $I \subset [n]$*

*consisting of semi-honest parties where at least one party is honest, while the rest are semi-honest. The protocol $\pi$ is considered to privately compute the Beaver multiplication $f(x,y)$ if a probabilistic polynomial-time simulator $S_i$ can be constructed such that, given all potential inputs, the simulated view of $P_i$ is statistically indistinguishable from $P_i$'s actual view during the execution of the protocol.*

*Proof.* The functionality of the protocol $\pi$, denoted as $f(x,y) := w$, is deterministic and ensures correctness. This correctness is verified by computing the sum of all outputs from $n$ parties and evaluating this sum modulo $\mathbb{Z}_{2^k}$, specifically, $(w_1 + w_2 + \ldots + w_n \bmod \mathbb{Z}_{2^k})$. The protocol specifies for each party $P_i$ a corresponding component of $f(x,y)$, $f_i(x,y) := w_i$, where $w_i$ is the output computed by party $P_i$.

Define the view of party $P_i$ during the execution of protocol $\pi$ as:

$$\text{view}_i^\pi(x,y) := (x_i, y_i, r_i, m_1, m_2, \ldots, m_t) \in \mathbb{Z}_{2^k},$$

where $r_i$ represents the outcome of the $P_i$ internal coin tosses and $m_1, m_2, \ldots, m_t$ denote the received messages.

To prove that $\pi$ privately computes $f$, we must show that there exists a probabilistic polynomial-time simulator $S$ for every $I$ such that:

$$\{S(x_I, f_I(x,y))\}_{x,y} \stackrel{\text{perf}}{\equiv} \{\text{view}_I^\pi(x,y)\}_{x,y}.$$

The actual view of $P$ in protocol $\pi$ for the multiplication function $f(x,y)$ is :

$$\text{view}_I^\pi(x,y) = (x_I, y_I, a_I, b_I, c_I, \delta_I, \epsilon_I)$$

where $y_1, \delta_I$, and $\epsilon_I$ are messages received by $P_I$ from other parties. Also, $a_I$, $b_I$ and $c_I$ are the Beaver triples received by $P_I$ for a multiplication.

It is important to note that all these elements are additively shared between the parties or results of the addition of two additively shared values. Therefore, based on the definition in Section B.1, they are independent uniformly random within the $\mathbb{Z}_{2^k}$ set.

Furthermore, from the $\text{view}_I^\pi(x,y)$, $P_I$ can compute all other relevant quantities and this view contains all of $P_I$'s information.

Now, we construct a simulator, denoted $S$, for the view of party $P_I$ that simulates the computation of $\delta$ and $\epsilon$, representing the differences between simulated inputs $\tilde{x}_I$ and $\tilde{y}_I$, and predetermined shares $\tilde{a}_I$ and $\tilde{b}_I$ from Beaver triples. These simulated values are independent and chosen uniformly at random, ensuring they mirror the actual protocol's input and share distribution. The simulator then calculates $\tilde{\delta}_I = \tilde{x}_I - \tilde{a}_I$ and $\tilde{\epsilon}_I = \tilde{y}_I - \tilde{b}_I$, mirroring the protocol's approach to determining $\delta$ and $\epsilon$.

With $\delta$ and $\epsilon$ computed, $S$ proceeds to simulate the computation of the share of the product, $\tilde{w}_I$, using the formula $\tilde{w}_I = \tilde{c}_I + \epsilon \cdot \tilde{b}_I + \delta \cdot \tilde{a}_I + \epsilon \cdot \delta$. This step effectively mimics the actual protocol's computation, incorporating the adjustment based on the computed differences ($\delta$ and $\epsilon$) and the correction factor ($\tilde{\epsilon}_I \cdot \tilde{\delta}_I$), utilizing the simulated shares of $a$, $b$, and $c$ from the Beaver triples.

We define simulator $S$ view for the multiplication as

$$S(x_I, f_I(x,y)) = (x, \tilde{x}_1, \tilde{y}_1, \tilde{a}_I, \tilde{b}_I, \tilde{c}_I, \tilde{\delta}_I \tilde{\epsilon}_I)$$

where each component of $S$ is independent uniformly random in $\mathbb{Z}_{2^k}$. Since $S$ selects and computes values that are independently and uniformly random, and from the same distribution as those in $\text{view}^\pi$, they are indistinguishable. As a result, the statistical distance between the simulated view $S(x_I, f_I(x,y))$ and the actual view $\text{view}_I^\pi(x,y)$ is effectively zero for all $x, y$. Consequently, all components, except for $x$, are indistinguishable from random values, thereby affirming the protocol's privacy. $\qquad\square$

**Theorem 4** (privacy w.r.t semi-honest behavior)**.** *Consider a secure multi-party computation protocol $\pi$ designed to compute a function $f(x)$ utilizing the inverse function (detailed in Section B.3). This*

*protocol involves $n$ parties, denoted as $P_1, P_2, \ldots, P_n$, with a subset $I \subset [n]$ consisting of semi-honest parties where at least one party is honest, while the rest are semi-honest. The protocol $\pi$ is considered to privately compute the inverse function $f(x)$ if a probabilistic polynomial-time simulator $S_i$ can be constructed such that, given all potential inputs, the simulated view of $P_i$ is statistically indistinguishable from $P_i$'s actual view during the execution of the protocol.*

*Proof.* The protocol $\pi$ ensures deterministic correctness by summing the outputs of $n$ parties modulo $\mathbb{Z}_{2^k}$. Each party $P_i$ contributes a specific output $w_i$ to the function $f(x, y)$. The privacy of $\pi$ is shown by proving that for any inputs $x$ and $y$, a simulator can generate a view for party $P$ that is indistinguishable from its actual view during the protocol's execution, encompassing inputs, internal randomness, and received messages.

The inversion function consists of three main operations on secret-shared inputs: addition, multiplication by a public constant, and Beaver multiplication. In contrast to Beaver multiplication, both the addition and the multiplication by a public constant operations are executed locally on the secret-shared data.

The inverse function begins with a "for" loop, within which there are two Beaver multiplications; the output of the first multiplication serves as the input for the second Beaver multiplication. Therefore, we can define the operations inside this loop as $f(x, y)$ and represent this function in the following form:

$$f'(x, y) = x \times y = w$$
$$f''(w, x) = w \times x$$

Here, $w$ is the output of the first multiplication $f'(x, y)$, and the output of $f''(w, x)$ is equal to $f(x, y)$.

Based on Theorem 3, protocol $\pi$ can compute each of the multiplication privately. As long as $w1$, $w2$, $\ldots$ and $w_n$, the shares of the Beaver multiplications output, are not revealed, they remain uniformly random and independent. However, since the output of the first multiplication is the input of the second multiplication, we need to show that the first multiplication $f'(x, y)$ remains private when the second multiplication $f''(w, x)$ is executed. In other words, the information received by $P$ during the second multiplication should not reveal any details about the inputs $x$ and $y$ of the first multiplication.

Therefore, we need to define the actual view of $P_I$ within protocol $\pi$ specifically for the second multiplication:

$$\begin{aligned}
\text{view}_I^\pi(w, x) = (&x_I, y_I, w_I, a_I, b_I, c_I, a'_I, b'_I, c'_I, \\
&\delta_2, \delta_3, \ldots, \delta_n, \epsilon_2, \epsilon_3, \ldots, \epsilon_n, \\
&\delta'_2, \delta'_3, \ldots, \delta'_n, \epsilon'_2, \epsilon'_3, \ldots, \epsilon'_n)
\end{aligned}$$

where $y_1$ is the share of initial guess, $\delta_2, \delta_3, \ldots, \delta_n$, and $\epsilon_2, \epsilon_3, \ldots, \epsilon_n$ are messages received by $P$ from other parties for the second multiplication, and $\delta'_2, \delta'_3, \ldots, \delta'_n, \epsilon'_2, \epsilon'_3, \ldots, \epsilon'_n$ are messages received by $P$ from other parties for the first multiplication. Also, $a_I$, $b_I$ and $c_I$ and $a'_I$, $b'_I$ and $c'_1$ are the Beaver triples received by $P$ for the multiplications and $w_1$ is the result of first multiplication.

All these elements, except for $x$, are additively shared between the parties or results of the addition of two additively shared values. Therefore, based on the definition in B.1, they are independent uniformly random within the $\mathbb{Z}_{2^k}$ set.

Furthermore, from the $\text{view}_I^\pi(w, x)$, $P$ can compute all other relevant quantities and this view contains all of $P$'s information.

Now, we construct a simulator, denoted $S$, for the view of party $P$ that simulates the computation of $\delta$ and $\epsilon$, representing the differences between simulated inputs $\tilde{x}_I$ and $\tilde{y}_I$, and predetermined shares $\tilde{a}_I$

and $\tilde{b}_I$ from Beaver triples. These simulated values are independent and chosen uniformly at random, ensuring they mirror the actual protocol's input and share distribution. The simulator then calculates $\tilde{\delta} = \tilde{x} - \tilde{a}_I$ and $\tilde{\epsilon} = \tilde{y} - \tilde{b}_I$, mirroring the protocol's approach to determining $\delta$ and $\epsilon$.

With $\delta$ and $\epsilon$ computed, $S$ proceeds to simulate the computation of the share of the product, $\tilde{w}_I$, using the formula $\tilde{w}_I = \tilde{c}_I + \epsilon \cdot \tilde{b}_I + \delta \cdot \tilde{a}_I + \epsilon \cdot \delta$. This step effectively mimics the actual protocol's computation, incorporating the adjustment based on the computed differences ($\delta$ and $\epsilon$) and the correction factor ($\tilde{\epsilon} \cdot \tilde{\delta}$), utilizing the simulated shares of $c$, $a$, and $b$ from the Beaver triples.

Now, we define simulator $S$ as

$$S(x_I, f_I(w, x)) = (\tilde{x}_I, \tilde{y}_I, \tilde{a}_I, \tilde{b}_I, \tilde{c}_I, \tilde{a}'_I, \tilde{b}'_I, \tilde{c}'_I, \tilde{\delta}_2$$
$$, \tilde{\delta}_3, \ldots, \tilde{\delta}_n, \tilde{\delta}'_2, \tilde{\delta}'_3, \ldots, \tilde{\delta}'_n, \tilde{\epsilon}_2, \tilde{\epsilon}_3$$
$$, \ldots, \tilde{\epsilon}_n, \tilde{\epsilon}'_2, \tilde{\epsilon}'_3, \ldots, \tilde{\epsilon}'_n)$$

Unlike the first beaver multiplication, the simulator view for the second multiplication consists of all messages $P$ receives during the second multiplication, plus what it could see from the first multiplication.

As shown, all components of $S$ are independent and uniformly random in $\mathbb{Z}_{2^k}$, except for $x$, which is $P$'s private input. Since $S$ selects and computed values that are independently and uniformly random, and from the same distribution as those in $\text{view}^\pi$, they are indistinguishable. As a result, the statistical distance between the simulated view $S(x, f''_I(w, x))$ and the actual view $\text{view}^\pi_I(w, x)$ is effectively zero for all $w, x$. Consequently, all components, except for $x$, are indistinguishable from random values, thereby affirming the protocol's privacy.

Moreover, since for each multiplication, only a new set of Beaver triples is used, the values $\tilde{\gamma}_n, \tilde{\delta}_n$ are different from $\tilde{\gamma}'_n, \tilde{\delta}'_n$ in the second multiplication. Therefore, all these intermediate value remain indistinguishable from random values.

Before discussing the actual and simulated views of the entire inversion function, we first need to demonstrate that the local operations on share values in inverse function do not reveal any information about the output share of the second multiplication. As shown in this algorithm, each client subtract its output share of the second multiplication (which we call $w'$) from twice of one of the input share of the multiplication, $x$. Therefore, we need to demonstrate that simulator $S$ can mimic the actual protocol's computation:

$$\llbracket w'' \rrbracket = 2 \times \llbracket x \rrbracket - \llbracket w' \rrbracket$$

To ensure the indistinguishability of the simulated computation from the actual protocol's operation $2 \times \llbracket x \rrbracket_i - \llbracket w' \rrbracket_i$, the simulator $S$ selects uniformly distributed and independent random variables, $\llbracket \tilde{x} \rrbracket$ and $\llbracket \tilde{w}' \rrbracket$. These variables are crucial for mimicking the actual computation without revealing any sensitive information about the output of the second multiplication, $w'$, or the input, $x$. By calculating $\llbracket \tilde{w}'' \rrbracket = 2 \times \llbracket \tilde{x} \rrbracket - \llbracket \tilde{w}' \rrbracket$ with these selected variables, $S$ manages to simulate the operation in a way that the outcome mirrors the distribution and independence characteristics of the actual computation.

The above steps represent a single execution of the loop inside the inversion function. To show the complete simulator view for this function, we need to demonstrate the simulator view when these steps are repeated $k$ times:

$$S(x_I, f_I(x)) = (\tilde{x}_I, \tilde{y}_I, \tilde{a}_I^{(1)}, \tilde{b}_I^{(1)}, \tilde{c}_I^{(1)}, \tilde{a}_I'^{(1)}, \tilde{b}_I'^{(1)}, \tilde{c}_I'^{(1)}, \tilde{\delta}_2^{(1)},$$
$$\tilde{\delta}_3^{(1)}, \ldots, \tilde{\delta}_n^{(1)}, \tilde{\delta}_2'^{(1)}, \tilde{\delta}_3'^{(1)}, \ldots, \tilde{\delta}_n'^{(1)}, \tilde{\epsilon}_2^{(1)}, \tilde{\epsilon}_3^{(1)},$$
$$\ldots, \tilde{\epsilon}_n^{(1)}, \tilde{\epsilon}_2'^{(1)}, \tilde{\epsilon}_3'^{(1)}, \ldots, \tilde{\epsilon}_n'^{(1)}, \ldots, \tilde{a}_1^{(k)}, \tilde{b}_1^{(k)},$$
$$\tilde{c}_1^{(k)}, \tilde{a}_1'^{(k)}, \tilde{b}_1'^{(k)}, \tilde{c}_1'^{(k)}, \tilde{\delta}_2^{(k)}, \tilde{\delta}_3^{(k)}, \ldots, \tilde{\delta}_n^{(k)},$$
$$\tilde{\delta}_2'^{(k)}, \tilde{\delta}_3'^{(k)}, \ldots, \tilde{\delta}_n'^{(k)}, \tilde{\epsilon}_2^{(k)}, \tilde{\epsilon}_3^{(k)}, \ldots, \tilde{\epsilon}_n^{(k)}, \tilde{\epsilon}_2'^{(k)},$$
$$\tilde{\epsilon}_3'^{(k)}, \ldots, \tilde{\epsilon}_n'^{(k)}, \tilde{w}^{(1)}, \tilde{w}'^{(1)}, \tilde{w}''^{(1)}, \ldots, \tilde{w}^{(k)},$$
$$\tilde{w}'^{(k)}, \tilde{w}''^{(k)})$$

Every component of $S$ is independent and uniformly random within $\mathbb{Z}_{2^k}$, with the exception of $x$, which represents $P$'s private input. As $S$ is designed to select values that are not only independently and uniformly random but also drawn from the same distribution as those in $\text{view}_I^\pi$, these components become indistinguishable. This indistinguishability, applicable to all components other than $x$, ensures that they resemble random values, thus reinforcing the privacy aspect of the protocol. $\square$

**Theorem 5** (privacy w.r.t semi-honest behavior). *Consider a secure multi-party computation protocol $\pi$ designed to compute a function $f(x)$ utilizing the square root function (detailed in Section B.4). This protocol involves $n$ parties, denoted as $P_1, P_2, \ldots, P_n$, with a subset $I \subset [n]$ consisting of semi-honest parties where at least one party is honest, while the rest are semi-honest. The protocol $\pi$ is considered to privately compute the square root function $f(x)$ if a probabilistic polynomial-time simulator $S_i$ can be constructed such that, given all potential inputs, the simulated view of $P_i$ is statistically indistinguishable from $P_i$'s actual view during the execution of the protocol.*

*Proof.* The protocol $\pi$ ensures deterministic correctness by summing the outputs of $n$ parties modulo $\mathbb{Z}_{2^k}$. Each party $P_i$ contributes a specific output $w_i$ to the function $f(x)$. The privacy of $\pi$ is shown by proving that for any input $x$, a simulator can generate a view for party $P$ that is indistinguishable from its actual view during the protocol's execution, encompassing inputs, internal randomness, and received messages.

The square root function consists of four main operations on secret-shared data: addition, multiplication by a public constant, Beaver multiplication and inversion. The addition and the multiplication by a public constant operations are executed locally on the secret-shared data.

The square root function begins with a 'for' loop that iterates $k$ times, with $k$ being fixed and known to all parties. Within this loop, the inverse of the initial guess is computed. Subsequently, the output of the inversion function is multiplied by the share of input of the square root function using Beaver multiplication. Following this, the output of this multiplication is locally added to the initial guess. Finally, the output from the previous computation is multiplied by 1/2 to yield the final approved result for the square root function.

We now define the actual view of $P$ within protocol $\pi$ for the square function:

$$
\begin{aligned}
\text{view}_I^\pi(x) = (x_I, m_2^{inv(1)}, m_3^{inv(1)}, \ldots, m_n^{inv(1)}, m_2^{mul(1)}, \\
m_3^{mul(1)}, \ldots, m_n^{mul(1)}, \ldots, m_2^{inv(k)}, m_3^{inv(k)} \\
, \ldots, m_n^{inv(k)}, m_2^{mul(k)}, m_3^{mul(k)}, \ldots, m_n^{mul(k)})
\end{aligned}
$$

where $x_I$ is a share of the input of square function, $m_1^{inv}, m_2^{inv}, \ldots,$ $m_n^{inv}$ are messages received by $P$ from other parties for the inversion operation and $m_2^{mul}, m_3^{mul}, \ldots, m_n^{mul})$ are messages received by $P$ from other parties for the Beaver multiplication operation in each of $k$ iterations.

All these elements, are additively shared between the parties or results of the addition of two additively shared values. Therefore, based on the definition in Algorithm 2.2, they are independent uniformly random within the $\mathbb{Z}_{2^k}$ set.

Furthermore, from the $\text{view}_I^\pi(x)$, $P$ can compute all other relevant quantities and this view contains all of $P$'s information.

Now, we construct a simulator, denoted $S$, for the view of party $P$ that simulates the computation of the square root function. The simulator $S$ begins by simulating the initial setup, including the distribution of the secret-shared input $\tilde{Y}_1$ and the initial guess $\tilde{x}_0$ for the square root computation. For the inversion operation, $S$ simulates the messages received from other parties during the execution of the inverse function $\tilde{m}_1^{inv}, \tilde{m}_2^{inv}, \ldots, \tilde{m}_n^{inv}$ and the outcome of the inversion function $\tilde{w}^{inv}$. According to Theorem 4, the inverse function can be computed privately, and the simulator can successfully simulate these exchanged messages, which are indistinguishable from random values in $\mathbb{Z}_{2^k}$.

For the Beaver multiplication, $S$ simulates the Beaver triples and the messages exchanged during the Beaver multiplication process $\tilde{m}_1^{mul}, \tilde{m}_2^{mul}, \ldots, \tilde{m}_n^{mul}$. In simulating the Beaver multiplication function, $S$ adheres to the privacy guarantees provided by Theorem 3. Therefore, the simulator can successfully simulate these exchanged messages, which are indistinguishable from random values in $\mathbb{Z}_{2^k}$. Also, since in actual protocol, parties do not reveal the shares of the output of Beaver multiplication function $\tilde{w}_n^{mul}$, $S$ can simulate $\tilde{w}_1^{mul}$ as uniformly random and independent values in $\mathbb{Z}_{2^k}$ that makes it indistinguishable.

The final steps of the algorithm involve local computations by $P$, including adding the result of Beaver multiplication (which is called here $[\![w']\!]$) to the current estimate of square root ( $[\![x_n]\!]$) and then multiplying by $1/2$ to adjust for the next iteration or final result. In this step, $S$ simulates these computations by following the same steps, the simulator $S$ selects uniformly distributed and independent random variables, $[\![\tilde{w}']\!]$ and $[\![\tilde{x_n}']\!]$ from $\mathbb{Z}_{2^k}$ and compute the final share value for the this round of approximation as follows:

$$[\![\tilde{x}_{n+1}]\!] \leftarrow \frac{1}{2}\left([\![\tilde{x}_n]\!] + [\![\tilde{w'}]\!]\right)$$

These steps effectively mimic the actual protocol's computation, and since the square root computation iterates this process $k$ times, $S$ repeats the simulation for each iteration, updating the simulated values according to the algorithm's steps.

Now we show the complete simulator view for square function:

$$\begin{aligned}
S(x_I, f_I(x)) = (x_I, &\tilde{m}_2^{inv(1)}, \tilde{m}_3^{inv(1)}, \ldots, \tilde{m}_n^{inv(1)}, \tilde{m}_2^{inv}, \\
&\tilde{m}_3^{inv(2)}, \ldots, \tilde{m}_n^{inv(2)}, \ldots, \tilde{m}_2^{inv(k)}, \\
&\tilde{m}_3^{inv(k)}, \ldots, \tilde{m}_n^{inv(k)}, \tilde{m}_2^{mul(1)}, \\
&\tilde{m}_3^{mul(1)}, \ldots, \tilde{m}_n^{mul(1)}, \tilde{m}_2^{mul(2)}, \\
&\tilde{m}_3^{mul(2)}, \ldots, \tilde{m}_n^{mul(2)}, \ldots, \tilde{m}_2^{mul(k)}, \\
&\tilde{m}_3^{mul(k)}, \ldots, \tilde{m}_n^{mul(k)}, \tilde{w}_1^{mul(1)}, \\
&\tilde{w}_1^{mul(2)}, \ldots, \tilde{w}_1^{mul(k)}, \tilde{w}_1^{inv(1)}, \\
&\tilde{w}_1^{inv(2)}, \ldots, \tilde{w}_1^{inv(k)})
\end{aligned}$$

Every component of $S$ is independent and uniformly random within $\mathbb{Z}_{2^k}$ which represents $P$'s private input. As $S$ is designed to select values that are not only independently and uniformly random but also drawn from the same distribution as those in $\text{view}_I^\pi$, these components become indistinguishable. This indistinguishability, applicable to all components ensures that they resemble random values, thus reinforcing the privacy aspect of the protocol. $\square$

**Theorem 6** (privacy w.r.t semi-honest behavior). *Consider a secure multi-party computation protocol $\pi$ designed to compute the norm of a vector, as detailed in line 1-3 of Algorithm 3. This protocol involves $n$ parties, denoted as $P_1, P_2, \ldots, P_n$, with a subset $I \subset [n]$ consisting of semi-honest parties where at least one party is honest, while the rest are semi-honest. Each party $P_i$ holds an additive share of a set of private inputs $x$, where $x$ represents the collective input vector shared among the parties. The protocol $\pi$ is said to privately compute the norm function $f(x)$ if there exists a probabilistic polynomial-time simulator $S$ that can simulate a view for any subset of parties that is statistically indistinguishable from their actual views during the protocol's execution, given all potential inputs.*

*Proof.* The protocol $\pi$ guarantees deterministic correctness by computing the sum of the outputs from $n$ parties modulo $\mathbb{Z}_{2^k}$, where each party $P_i$ contributes an output $w_i$ corresponding to their share of the computation of $f(x)$. To demonstrate the privacy of $\pi$, we show that for any collective input $x$, a simulator $S$ can generate a simulated view for any honest party or group of honest parties that mirrors their actual experience during the protocol's execution. This includes their inputs, the internal randomness used, and the messages received from other parties.

The norm computation involves three key operations on secret-shared data: Beaver multiplication for squaring the input vector, local summation of the squared values, and finally, square root computation to derive the norm.

The actual view of $P_I$, for the computation is as follows:

$$\text{view}_I^\pi(x) = (x_I, m_1^{mul}, m_2^{mul}, \ldots, m_n^{mul}, m_1^{sqrt}, m_2^{sqrt},$$
$$\ldots, m_n^{sqrt})$$

where $x_I$ denotes the vector of additive shares of the input held by all honest parties, $m_i^{mul}$ are the messages exchanged during the Beaver multiplication, and $m_i^{sqrt}$ are the messages exchanged for the square root computation.

Given the additive sharing scheme, all elements involved in $\text{view}_I^\pi(\vec{x})$ are independently and uniformly distributed within $\mathbb{Z}_{2^k}$, ensuring that from $P$'s perspective, all components are indistinguishable from random. This setup, based on the privacy-preserving properties of Beaver multiplication and square root operations, allows $S$ to simulate a plausible set of interactions that $P$ (or any group of honest parties) would observe, without revealing any private information. The simulator $S$ thus constructs a simulated view that includes simulated messages for each step of the norm computation, ensuring these simulated components are indistinguishable from those in a real protocol execution.

We construct a simulator, denoted $S$, for simulating the collective view of honest parties in the vector norm computation within the **SecureDL** protocol. The role of $S$ is to create a plausible simulation of the protocol execution as it would appear to the honest parties, focusing on the data exchanges and computations that occur without assuming control over the adversary's direct observations.

In the case of Beaver multiplication, $S$ simulates the Beaver triples and the messages exchanged during this phase, such as $\tilde{m}_1^{mul}, \tilde{m}_2^{mul}, \ldots, \tilde{m}_n^{mul}$. This simulation is consistent with the privacy assurances specified in Theorem 3, making these simulated messages indistinguishable from random values in $\mathbb{Z}_{2^k}$ to any observer, thereby maintaining the integrity of the privacy guarantees.

For local computations, including the summation of outputs, the simulator $S$ performs:

$$[\![\tilde{\text{sum}}]\!] \leftarrow \sum_{i=1}^{n} [\![\tilde{w}_i^{mul}]\!],$$

imitating the aggregation process faithfully without disclosing any individual's private data.

During the square root computation, $S$ constructs messages such as $\tilde{m}_1^{sqrt}, \tilde{m}_2^{sqrt}, \ldots, \tilde{m}_n^{sqrt}$, simulating the inter-party communications. These simulated messages are crafted to be statistically indistinguishable from those in a genuine protocol run, aligning with the privacy-preserving methodology of the square root function described in Theorem 5.

Considering multiple honest parties, the simulated view $S$ provides for the protocol includes the inputs of all such parties, denoted as $x_1, x_2, \ldots, x_m$, where $m$ is the number of honest parties. Thus, the complete simulator view is represented as:

$$S(x_I, f_I(x)) = (x_I, \tilde{m}_1^{mul}, \tilde{m}_2^{mul}, \ldots, \tilde{m}_n^{mul}, \tilde{m}_1^{sqrt}, \tilde{m}_2^{sqrt},$$
$$\ldots, \tilde{m}_n^{sqrt})$$

Each component of $S$ is independently and uniformly random within $\mathbb{Z}_{2^k}$, ensuring that the simulation reflects the private inputs and the exchange of messages among honest parties, all while remaining indistinguishable from a real execution to any external observer. $\qquad\square$

**Theorem 7** (privacy w.r.t semi-honest behavior). *Consider a secure multi-party computation protocol $\pi$ designed to compute a function $f(x, y)$, utilizing the cosine similarity computation between two vectors (as detailed in Section 5). This protocol involves $n$ parties, denoted as $P_1, P_2, \ldots, P_n$, with a subset $I \subset [n]$ consisting of semi-honest parties where at least one party is honest, while the rest may*

*be semi-honest. Each party $P_i$ possesses an additive share of the private inputs $x$ and $y$. The protocol $\pi$ is considered to effectively compute the cosine similarity function $f(x,y)$ in a private manner if a probabilistic polynomial-time simulator $S$ can be constructed. For any given set of potential inputs, the simulated view of $P_i$ should be statistically indistinguishable from $P_i$'s actual view during the execution of the protocol.*

*Proof.* The protocol $\pi$ ensures deterministic correctness by summing the outputs of $n$ parties modulo $\mathbb{Z}_{2^k}$. Each party $P_i$ contributes a specific output $w_i$ to the function $f(x,y)$. The privacy of $\pi$ is shown by proving that for any inputs $x$ and $y$, a simulator can generate a view for party $P$ that is indistinguishable from its actual view during the protocol's execution, encompassing inputs, internal randomness, and received messages.

In the first step of the cosine similarity computation function, the dot product of the inputs should be computed. In our work, a single Beaver multiplication is utilized to perform this operation. Following this, the norm of each vector is computed, and then these norm values are multiplied using Beaver triple multiplication. Subsequently, the inverse of this multiplication needs to be computed. Finally, the result of that last step is multiplied by the dot product results computed in the first step to calculate the final output of the cosine similarity function.

We now define the actual view of $P$ within protocol $\pi$ for the square function:

$$
\begin{aligned}
\text{view}_I^\pi(x,y) = (x_I, y_I m_1^{mul_1}, m_2^{mul_1}, \ldots, m_n^{mul_1}, m_1^{norm_1}, \\
m_2^{norm_1}, \ldots, m_n^{norm_1}, m_1^{norm_2}, m_2^{norm_2}, \ldots, \\
m_n^{norm_2}, m_1^{mul_2}, m_2^{mul_2}, \ldots, m_n^{mul_2}, m_1^{inv}, \\
m_2^{inv}, \ldots, m_n^{inv}, m_1^{mul_3}, m_2^{mul_3}, \ldots, m_n^{mul_3})
\end{aligned}
$$

where $x_I$ and $y_I$ are the share of the inputs of norm computation function, $m_1^{mul_1}, m_2^{mul_1}, \ldots, m_n^{mul_1}$, $m_1^{mul_2}, m_2^{mul_2}, \ldots, m_n^{mul_2}$ and $m_1^{mul_3}, m_2^{mul_3}, \ldots, m_n^{mul_3}$ are messages received by $P$ from other parties for the Beaver multiplication operations and $m_1^{norm_1}, m_2^{norm_1}$, $\ldots, m_n^{norm_1}$ and $m_1^{norm_2}, m_2^{norm_2}, \ldots, m_n^{norm_2}$ are messages received by $P$ from other parties for the norm computation operation and finally $m_1^{inv}, m_2^{inv}, \ldots, m_n^{inv}$ are messages received by $P$ from other parties for the inversion operation.

All these elements, are additively shared between the parties or results of the addition of two additively shared values. Therefore, based on the definition in Algorithm 2.2, they are independent uniformly random within the $\mathbb{Z}_{2^k}$ set. Furthermore, from the $\text{view}_I^\pi(x,y)$, $P$ can compute all other relevant quantities and this view contains all of $P$'s information.

Now, we construct a simulator, denoted $S$, for the view of party $P$ that simulates the computation of the cosine similarity computation function. The simulator $S$ begins by simulating the initial setup, including the of the secret-shared inputs $\tilde{w}_a$ and $\tilde{w}_b$. For the Beaver multiplications, $S$ simulates the Beaver triples and the messages exchanged during the Beaver multiplication process $\tilde{m}_1^{mul_1}, \tilde{m}_2^{mul_1}, \ldots, \tilde{m}_n^{mul_1}, \tilde{m}_1^{mul_2}, \tilde{m}_2^{mul_2}, \ldots, \tilde{m}_n^{mul_2}$. In simulating the Beaver multiplication function, $S$ adheres to the privacy guarantees provided by Theorem 3. Therefore, the simulator can successfully simulate these exchanged messages, which are indistinguishable from random values in $\mathbb{Z}_{2^k}$. Also, since in actual protocol, parties do not reveal the shares of the output of Beaver multiplication functions in any steps $\tilde{w}_n^{mul_1}, \tilde{w}_n^{mul_2}$ and $\tilde{w}_n^{mul_3}$, $S$ can simulate them as uniformly random and independent values in $\mathbb{Z}_{2^k}$ that makes it indistinguishable.

Next step involves vector norm computations, $S$ simulates the messages exchanged during the computation process $\tilde{m}_1^{norm_1}, \tilde{m}_2^{norm_1}, \ldots, \tilde{m}_n^{norm_1}$ and $\tilde{m}_1^{norm_2}, \tilde{m}_2^{norm_2}, \ldots, \tilde{m}_n^{norm_2}$. In simulating the norm computation function, $S$ adheres to the privacy guarantees provided by Theorem 6. Therefore, the simulator can successfully simulate these exchanged messages, which are indistinguishable from random values in $\mathbb{Z}_{2^k}$. Also, since in actual protocol, parties do not reveal the shares of the output of norm functions in any steps $\tilde{w}_n^{norm_1}$ and $\tilde{w}_n^{norm_2}$, $S$ can simulate them as uniformly random and independent values in $\mathbb{Z}_{2^k}$ that makes it indistinguishable.

Finally, the inversion computation operation, $S$ simulates the messages received from other parties during the execution of this function $\tilde{m}_1^{inv}, \tilde{m}_2^{inv}, \ldots, \tilde{m}_n^{inv}$ and the outcome of the inversion function $\tilde{w}^{inv}$. According to Theorem 4, the inversion function can be computed privately, and the simulator can successfully simulate these exchanged messages, which are indistinguishable from random values in $\mathbb{Z}_{2^k}$. Also, since in actual protocol, parties do not reveal the shares of the output of inversion function $\tilde{w}_n^{inv}$, $S$ can simulate $\tilde{w}_1^{inv}$ as uniformly random and independent values in $\mathbb{Z}_{2^k}$ that makes it indistinguishable.

Now we show the complete simulator view for cosine similarity computation:

$$
\begin{aligned}
S(x_I, f_I(x,y)) = (x_I, y_I, \tilde{m}_1^{mul_1}, \tilde{m}_2^{mul_1}, \ldots, \tilde{m}_n^{mul_1}, \tilde{m}_1^{mul_2} \\
, \tilde{m}_2^{mul_2}, \ldots, \tilde{m}_n^{mul_2} \, \tilde{m}_1^{mul_3}, \tilde{m}_2^{mul_3}, \ldots, \\
\tilde{m}_n^{mul_3} \, \tilde{m}_1^{norm_1}, \tilde{m}_2^{norm_1}, \ldots, \tilde{m}_n^{norm_1}, \\
\tilde{m}_1^{norm_2}, \tilde{m}_2^{norm_2}, \ldots, \tilde{m}_n^{norm_2}, \tilde{m}_1^{inv}, \tilde{m}_2^{inv} \\
, \ldots, \tilde{m}_n^{inv})
\end{aligned}
$$

Every component of $S$ is independent and uniformly random within $\mathbb{Z}_{2^k}$ which represents $P$'s private input. As $S$ is designed to select values that are not only independently and uniformly random but also drawn from the same distribution as those in $\text{view}_I^\pi$, these components become indistinguishable. This indistinguishability, applicable to all components ensures that they resemble random values, thus reinforcing the privacy aspect of the protocol. $\square$

**Theorem 8** (privacy w.r.t semi-honest behavior). *Consider a secure multi-party computation protocol $\pi$ designed to compute a function $f(x,y)$ utilizing the L2 normalization computation between two vectors (detailed in Section 3). This protocol involves $n$ parties, denoted as $P_1, P_2, \ldots, P_n$, with a subset $I \subset [n]$ consisting of semi-honest parties where at least one party is honest, while the rest are semi-honest. The protocol $\pi$ is considered to privately compute the cosine similarity function $f(x,y)$ if a probabilistic polynomial-time simulator $S_i$ can be constructed such that, given all potential inputs, the simulated view of $P_i$ is statistically indistinguishable from $P_i$'s actual view during the execution of the protocol.*

*Proof.* The protocol $\pi$ ensures deterministic correctness by summing the outputs of $n$ parties modulo $\mathbb{Z}_{2^k}$. Each party $P_i$ contributes a specific output $w_i$ to the function $f(x,y)$. The privacy of $\pi$ is shown by proving that for any inputs $x$ and $y$, a simulator can generate a view for party $P$ that is indistinguishable from its actual view during the protocol's execution, encompassing inputs, internal randomness, and received messages.

In the first step of the L2 normalization function, the norm of each input vector is computed. Then, the inverse of $[\![\mathbf{w}_a]\!]$, the vector to be normalized, is computed. Following this, the inverse value is multiplied by the norm of the reference vector $[\![\mathbf{w}_b]\!]$ using the Beaver multiplication function. Finally, the output of this multiplication is multiplied by $[\![\mathbf{w}_a]\!]$ using the Beaver multiplication function to produce the final output.

We now define the actual view of $P$ within protocol $\pi$ for the square function:

$$
\begin{aligned}
\text{view}_I^\pi(x,y) = (x_I, y_I m_1^{mul_1}, m_2^{mul_1}, \ldots, m_n^{mul_1}, m_1^{norm_1}, \\
m_2^{norm_1}, \ldots, m_n^{norm_1}, m_1^{norm_2}, m_2^{norm_2}, \ldots, \\
m_n^{norm_2}, m_1^{mul_2}, m_2^{mul_2}, \ldots, m_n^{mul_2}, m_1^{inv}, \\
m_2^{inv}, \ldots, m_n^{inv})
\end{aligned}
$$

where $x_I$ is a share of the input of norm computation function, $m_1^{mul_1}, m_2^{mul_1}, \ldots, m_n^{mul_1}$, and $m_1^{mul_2}, m_2^{mul_2},$ $\ldots, m_n^{mul_2}$ are messages received by $P$ from other parties for the Beaver multiplication operations and $m_1^{norm_1}$,

$m_2^{norm_1}, \ldots, m_n^{norm_1}$ and $m_1^{norm_2}, m_2^{norm_2}, \ldots, m_n^{norm_2}$ are messages received by $P$ from other parties for the norm computation operation and finally $m_1^{inv}, m_2^{inv}, \ldots, m_n^{inv}$ are messages received by $P$ from other parties for the inversion operation.

All these elements, are additively shared between the parties or results of the addition of two additively shared values. Therefore, based on the definition in Algorithm 2.2, they are independent uniformly random within the $\mathbb{Z}_{2^k}$ set. Furthermore, from the $\text{view}^\pi(x, y)$, $P$ can compute all other relevant quantities and this view contains all of $P$'s information.

Now, we construct a simulator, denoted $S$, for the view of party $P$ that simulates the computation of the cosine similarity computation function. The simulator $S$ begins by simulating the initial setup, including the of the secret-shared inputs $\tilde{w}_a$ and $\tilde{w}_b$. For the Beaver multiplications, $S$ simulates the Beaver triples and the messages exchanged during the Beaver multiplication process $\tilde{m}_1^{mul_1}, \tilde{m}_2^{mul_1}, \ldots, \tilde{m}_n^{mul_1}$ and $\tilde{m}_1^{mul_2}, \tilde{m}_2^{mul_2}, \ldots, \tilde{m}_n^{mul_2}$. In simulating the Beaver multiplication function, $S$ adheres to the privacy guarantees provided by Theorem 3. Therefore, the simulator can successfully simulate these exchanged messages, which are indistinguishable from random values in $\mathbb{Z}_{2^k}$. Also, since in actual protocol, parties do not reveal the shares of the output of Beaver multiplication functions in any steps $\tilde{w}_n^{mul_1}, \tilde{w}_n^{mul_2}$ and $S$ can simulate them as uniformly random and independent values in $\mathbb{Z}_{2^k}$ that makes it indistinguishable.

Next step involves vector norm computations, $S$ simulates the messages exchanged during the computation process $\tilde{m}_1^{norm_1}, \tilde{m}_2^{norm_1}, \ldots, \tilde{m}_n^{norm_1}$ and $\tilde{m}_1^{norm_2}, \tilde{m}_2^{norm_2}, \ldots, \tilde{m}_n^{norm_2}$. In simulating the norm computation function, $S$ adheres to the privacy guarantees provided by Theorem 6. Therefore, the simulator can successfully simulate these exchanged messages, which are indistinguishable from random values in $\mathbb{Z}_{2^k}$. Also, since in actual protocol, parties do not reveal the shares of the output of norm functions in any steps $\tilde{w}_n^{norm_1}$ and $\tilde{w}_n^{norm_2}$, $S$ can simulate them as uniformly random and independent values in $\mathbb{Z}_{2^k}$ that makes it indistinguishable.

Finally, the inversion computation operation, $S$ simulates the messages received from other parties during the execution of this function $\tilde{m}_1^{inv}, \tilde{m}_2^{inv}, \ldots, \tilde{m}_n^{inv}$ and the outcome of the inversion function $\tilde{w}^{inv}$. According to Theorem 4, the inversion function can be computed privately, and the simulator can successfully simulate these exchanged messages, which are indistinguishable from random values in $\mathbb{Z}_{2^k}$. Also, since in actual protocol, parties do not reveal the shares of the output of square root function $\tilde{w}_n^{inv}$, $S$ can simulate $\tilde{w}_1^{inv}$ as uniformly random and independent values in $\mathbb{Z}_{2^k}$ that makes it indistinguishable.

Now we show the complete simulator view for cosine similarity function:

$$
\begin{aligned}
S(x_I, f_I(x, y)) = (&x_1, y_1, \tilde{m}_1^{mul_1}, \tilde{m}_2^{mul_1}, \ldots, \tilde{m}_n^{mul_1}, \tilde{m}_1^{mul_2} \\
&, \tilde{m}_2^{mul_2}, \ldots, \tilde{m}_n^{mul_2} \, \tilde{m}_1^{norm_1}, \tilde{m}_2^{norm_1}, \ldots, \\
&\tilde{m}_n^{norm_1}, \tilde{m}_1^{norm_2}, \tilde{m}_2^{norm_2}, \ldots, \tilde{m}_n^{norm_2}, \\
&\tilde{m}_1^{inv}, \tilde{m}_2^{inv}, \ldots, \tilde{m}_n^{inv})
\end{aligned}
$$

Every component of $S$ is independent and uniformly random within $\mathbb{Z}_{2^k}$ which represents $P$'s private input. As $S$ is designed to select values that are not only independently and uniformly random but also drawn from the same distribution as those in $\text{view}_I^\pi$, these components become indistinguishable. This indistinguishability, applicable to all components ensures that they resemble random values, thus reinforcing the privacy aspect of the protocol. □

**Theorem 9** (privacy w.r.t semi-honest behavior). *Consider a secure multi-party computation protocol $\pi$ designed to compute a function $f(\mathbf{x}, \mathbf{y})$ that yields the Euclidean distance between two vectors, as detailed in Algorithm 4. This protocol involves $n$ parties, denoted as $P_1, P_2, \ldots, P_n$, with a subset $I \subset [n]$ consisting of semi-honest parties where at least one party is honest, while the rest are semi-honest. Each party $P_i$ holds additive shares of the private input vectors $\mathbf{x}$ and $\mathbf{y}$. The protocol $\pi$ is considered to privately compute the Euclidean distance function $f(\mathbf{x}, \mathbf{y})$ if a probabilistic polynomial-time simulator $S$ can be constructed such that, given all potential inputs, the simulated view of $P_i$ is statistically indistinguishable from $P_i$'s actual view during the execution of the protocol.*

*Proof.* The protocol $\pi$ guarantees deterministic correctness by computing the sum of outputs from $n$ parties modulo $\mathbb{Z}_{2^k}$, with each party $P_i$ contributing an output $w_i$ corresponding to their share in the computation of $f(\mathbf{x}, \mathbf{y})$. To prove privacy, we show that, for any input vectors $\mathbf{x}, \mathbf{y}$, a simulator $S$ can generate a view for any honest party or group of honest parties that is indistinguishable from their actual protocol view, including their inputs, internal randomness, and messages received from other parties.

The computation of the Euclidean distance between two secret-shared vectors proceeds as follows. Each party first computes the element-wise difference $[\![\mathbf{d}]\!] = [\![\mathbf{x}]\!] - [\![\mathbf{y}]\!]$ locally. Then, the element-wise square $[\![\mathbf{d}^2]\!] = [\![\mathbf{d}]\!] \times [\![\mathbf{d}]\!]$ is computed using Beaver multiplication (as per Theorem 3). Next, the parties locally sum all squared components to obtain $[\![\text{sum}_d]\!] = \sum_{i=1}^{m} [\![d_i^2]\!]$. Finally, the square root is securely computed as $[\![d_{xy}]\!] = \text{ComputeSquareRoot}([\![\text{sum}_d]\!])$ using the secure square root protocol (see Theorem 5).

The actual view of a subset $I$ of parties in protocol $\pi$ during the execution of the Euclidean distance function can thus be written as:

$$\text{view}_I^\pi(\mathbf{x}, \mathbf{y}) = (x_I, y_I, m_1^{mul}, m_2^{mul}, \ldots, m_n^{mul},$$
$$m_1^{sqrt}, m_2^{sqrt}, \ldots, m_n^{sqrt})$$

where $x_I$ and $y_I$ are the additive shares of $\mathbf{x}$ and $\mathbf{y}$ held by the honest parties, $m_i^{mul}$ are the messages exchanged during the Beaver multiplication for squaring, and $m_i^{sqrt}$ are the messages exchanged for the secure square root computation.

All these elements, apart from the input shares, are additively shared between the parties or derived from the addition or multiplication of additively shared values. According to the properties established in Algorithm 2.2, all these elements are independently and uniformly distributed within $\mathbb{Z}_{2^k}$.

We now construct a simulator $S$ for the view of a subset $I$ of honest parties. The simulator $S$ receives as input the shares $x_I$ and $y_I$, and generates a simulated view as follows. For the Beaver multiplication step, $S$ simulates the Beaver triples and the corresponding messages $\tilde{m}_1^{mul}, \tilde{m}_2^{mul}, \ldots, \tilde{m}_n^{mul}$ in accordance with the privacy guarantees of Theorem 3, ensuring that these simulated messages are independently and uniformly random within $\mathbb{Z}_{2^k}$ and thus indistinguishable from the real protocol messages. For the local summation, $S$ computes the sum of the simulated squared shares in the same way as the actual protocol, mirroring the aggregation step without leaking any additional information. For the square root computation, $S$ simulates the messages $\tilde{m}_1^{sqrt}, \tilde{m}_2^{sqrt}, \ldots, \tilde{m}_n^{sqrt}$ exchanged during the secure square root protocol as described in Theorem 5, with these simulated messages also drawn independently and uniformly at random.

Thus, the simulator's view can be written as:

$$S(x_I, y_I, f_I(\mathbf{x}, \mathbf{y})) = (x_I, y_I, \tilde{m}_1^{mul}, \tilde{m}_2^{mul}, \ldots, \tilde{m}_n^{mul},$$
$$\tilde{m}_1^{sqrt}, \tilde{m}_2^{sqrt}, \ldots, \tilde{m}_n^{sqrt})$$

where every component of $S$ is independently and uniformly random within $\mathbb{Z}_{2^k}$, except for the input shares, which are the private inputs of the parties.

Since $S$ selects values that are independently and uniformly random, and from the same distribution as those in $\text{view}_I^\pi(\mathbf{x}, \mathbf{y})$, the two views are statistically indistinguishable. This indistinguishability ensures that the simulated view is as secure as a real execution, upholding the privacy of the protocol.

Therefore, the protocol $\pi$ privately computes the Euclidean distance function $f(\mathbf{x}, \mathbf{y})$ in the presence of up to $n - 1$ semi-honest adversaries. $\qquad\square$

## APPENDIX I   ALGORITHMS

This appendix lists the auxiliary algorithms that are invoked throughout the main text. They are included here in full for clarity and reproducibility, covering the decentralized training loop as well as the secure subroutines (normalization, distance, and similarity computations) used in our framework.

---

**Algorithm 2:** Decentralized Learning Protocol

**Input** : Initial model parameters $w_c^0$ for $c \in C$

Local training data of the client $c$: $X_c$ for $c \in C$

**1 for** $t \in [0, 1, \ldots]$ **do**

**2**    **Local optimization step:**

**3**    **for** $c \in C$ **do**

**4**      Sample $x_c^t$ from $X_c$

**5**      Update parameters: $w_c^{t+\frac{1}{2}} = w_c^t - \eta \nabla_{w_c^t}(x_c^t, w_c^t)$

**6**    **Communication with neighbors:**

**7**    **for** $c \in C$ **do**

**8**      **for** $u \in \mathbf{N}(c) \setminus \{c\}$ **do**

**9**        Send $w_c^{t+\frac{1}{2}}$ to $u$

**10**        Receive $w_c^{t+\frac{1}{2}}$ from $u$

**11**    **Model updates aggregation:**

**12**    **for** $c \in C$ **do**

**13**      $w_c^{t+1} = \frac{1}{|\mathbf{N}(c)|} \sum_{c \in \mathbf{N}(c)} w_c^{t+\frac{1}{2}}$

---

**Algorithm 3:** ComputeL2Normalization

**Input** : Secret-shared vectors $[\![\mathbf{w}_a]\!]$ and $[\![\mathbf{w}_b]\!]$.

**Output** : L2-normalized vector $[\![\mathbf{w}_a']\!]$ relative to $[\![\mathbf{w}_b]\!]$.

  // Vector norm computation

**1** $[\![\mathbf{w}_a^2]\!] \leftarrow [\![\mathbf{w}_a]\!] \times [\![\mathbf{w}_a]\!]$

**2** $[\![\text{sum}_a]\!] \leftarrow \sum_{i=1}^n [\![w_{ai}^2]\!]$

**3** $[\![\|\mathbf{w}_a\|_2]\!] \leftarrow \text{ComputeSquareRoot}([\![\text{sum}_a]\!])$

  // Vector norm computation

**4** $[\![\mathbf{w}_b^2]\!] \leftarrow [\![\mathbf{w}_b]\!] \times [\![\mathbf{w}_b]\!]$

**5** $[\![\text{sum}_b]\!] \leftarrow \sum_{i=1}^n [\![w_{bi}^2]\!]$

**6** $[\![\|\mathbf{w}_b\|_2]\!] \leftarrow \text{ComputeSquareRoot}([\![\text{sum}_b]\!])$

**7** $[\![\|\mathbf{w}_a\|_2^{-1}]\!] \leftarrow \text{ComputeInverse}([\![\|\mathbf{w}_a\|_2]\!])$

  // Compute the ratio for normalization

**8** $[\![r]\!] \leftarrow [\![\|\mathbf{w}_b\|_2]\!] \times [\![\|\mathbf{w}_a\|_2^{-1}]\!]$

  // Normalize the vector $[\![\mathbf{w}_a]\!]$ by the computed ratio

**9** $[\![\mathbf{w}_a']\!] \leftarrow [\![\mathbf{w}_a]\!] \times [\![r]\!]$

**10 return** $[\![\mathbf{w}_a']\!]$

---

**Algorithm 4:** Secure Euclidean Distance Computation

**Input** : Secret-shared vectors $[\![\mathbf{w}_a]\!]$ and $[\![\mathbf{w}_b]\!]$, each of length $m$.

**Output** : Secret-shared Euclidean distance $[\![d_{ab}]\!]$ between $[\![\mathbf{w}_a]\!]$ and $[\![\mathbf{w}_b]\!]$.

  // Compute the element-wise difference

**1** $[\![\mathbf{d}]\!] \leftarrow [\![\mathbf{w}_a]\!] - [\![\mathbf{w}_b]\!]$

  // Element-wise squaring

**2 for** $i = 1$ **to** $m$ **do**

**3**    $[\![d_i^2]\!] \leftarrow [\![d_i]\!] \times [\![d_i]\!]$

  // Sum the squared differences

**4** $[\![\text{sum}_d]\!] \leftarrow \sum_{i=1}^m [\![d_i^2]\!]$

  // Compute the square root of the sum

**5** $[\![d_{ab}]\!] \leftarrow \text{ComputeSquareRoot}([\![\text{sum}_d]\!])$

**6 return** $[\![d_{ab}]\!]$

---

---

**Algorithm 5:** Cosine Similarity Computation

**Input** : Two secret-shared vectors $[\![\mathbf{w}_a]\!]$ and $[\![\mathbf{w}_b]\!]$.

**Output :** The cosine similarity $[\![\text{cosine}]\!]$ between the vectors $[\![\mathbf{w}_a]\!]$ and $[\![\mathbf{w}_b]\!]$.

1 $[\![\text{dotProduct}]\!] \leftarrow [\![\mathbf{w}_a]\!] \odot [\![\mathbf{w}_b]\!]$            // Vector dot product

  // Compute the norm of vector $\mathbf{w}_a$ using vectorized operations

2 $[\![\mathbf{w}_a^2]\!] \leftarrow [\![\mathbf{w}_a]\!] \times [\![\mathbf{w}_a]\!]$

3 $[\![\text{sum}_a]\!] \leftarrow \sum_{i=1}^{n} [\![w_{ai}^2]\!]$

4 $[\![\text{norm}_a]\!] \leftarrow \text{ComputeSquareRoot}([\![\text{sum}_a]\!])$

  // Compute the norm of vector $\mathbf{w}_b$ using vectorized operations

5 $[\![\mathbf{w}_b^2]\!] \leftarrow [\![\mathbf{w}_b]\!] \times [\![\mathbf{w}_b]\!]$

6 $[\![\text{sum}_b]\!] \leftarrow \sum_{i=1}^{n} [\![w_{bi}^2]\!]$

7 $[\![\text{norm}_b]\!] \leftarrow \text{ComputeSquareRoot}([\![\text{sum}_b]\!])$

8 $[\![\text{denom}]\!] \leftarrow \text{ComputeInverse}([\![\text{norm}_a]\!] \times [\![\text{norm}_b]\!])$       // Element-wise multiplication for norms and then inversion

9 $[\![\text{cosine}]\!] \leftarrow [\![\text{dotProduct}]\!] \times [\![\text{denom}]\!]$

10 **return** $[\![cosine]\!]$

---