# OpenReview forum: "SecureDL: Byzantine-Robust and Privacy-Preserving Aggregation for Decentralized Learning"
_ICLR.cc/2026/Conference — ICLR 2026 Conference Withdrawn Submission_

### Official Review · Reviewer_Yx7X · 2025-10-27

**Soundness:** 3
**Presentation:** 3
**Contribution:** 2
**Rating:** 4
**Confidence:** 4

**Summary:**

This paper proposes a novel, robust, and privacy-preserving protocol for decentralized learning paradigm. The proposed protocol is robust against model poisoning attacks by malicious users and ensures model privacy through an additive secret sharing scheme via two semi-honest non-colluding users. Using secret sharing as the core secure multi-party computation (MPC) primitive, proposed protocol integrates operations including cosine computation, normalization, scalar comparison, and weighted sums to achieve robust aggregation. Theoretical proofs are provided for demonstrating its convergence and privacy guarantees and experimental results are presented for evaluating the computational overhead.

**Strengths:**

This work is highlighted for the first to propose robust and privacy-preserving protocol via secret sharing scheme in the topic of fully decentralized learning, which excludes federated learning. The paper demonstrates strong originality and is clearly articulated. The technical claims are supported by rigorous privacy and convergence proofs.

**Weaknesses:**

This work claims to be the first to present a robust and privacy-preserving protocol, but similar protocols have been extensively studied in a boarder sense of decentralized learning, known as federated learning. This raises fundamental questions about the practical importance and utility of the DL paradigm, and specifically, why it offers superior benefits compared to established federated learning approaches. That is, how DL is important as a learning paradigm and why one should choose DL rather than FL. Though proofs are provided, they are standard using convergence analysis and simulator-based proof. Hence, the novelty of this part is limited. Furthermore, the protocol essentially combines existing methods, using secret sharing to integrate norms, cosine similarity, and scoring, making it a straightforward composition rather than a significant innovation.

In introduction, authors introduce DL as a scalable and communication-efficient learning paradigm, where participated clients can be resource-constrained devices. However, it seems the overhead is extremely high, from both DL’s nature and SS’s practice. Each user needs to communicate with multiple neighbors and encoding/decoding for MPC needs to be executed. This concern is exacerbated by the experimental setup. Authors present the configuration using high-performance computing cluster equipped with modern GPUs and multi-core CPUs without any specific specs. Yet, the datasets and models used are relatively naïve, such as MNIST, CIFAR10 with MLP, a shallow CNN. This mismatch undermines the claims of practical efficiency. When using a modern (up-to-date) model, such as transformer, will the proposed method remain scalable and practical?

**Questions:**

See Weakness.

---

### Official Review · Reviewer_hLYm · 2025-10-31

**Soundness:** 2
**Presentation:** 3
**Contribution:** 1
**Rating:** 2
**Confidence:** 4

**Summary:**

The paper proposes SecureDL, a decentralized learning framework aiming to provide Byzantine robustness and privacy preservation simultaneously. The approach integrates authenticated secret sharing (MAC-based MPC) to protect model updates, while leveraging cosine similarity and Euclidean distance metrics to identify and mitigate adversarial contributions during aggregation. The authors evaluate their method on multiple datasets (MNIST, CIFAR-10, SVHN, Fashion-MNIST), showing competitive performance and reasonable computational overhead compared to several existing robust aggregation baselines (e.g., DKrum, BRIDGE, UBAR, BALANCE).

**Strengths:**

1. The paper addresses an important and challenging problem at the intersection of Byzantine robustness and privacy preservation in decentralized learning.
2. The methodology is clearly described and the algorithms are well formalized.
3. The experimental section is relatively comprehensive, covering both IID and non-IID data distributions and various attack types.

**Weaknesses:**

1. Limited Novelty in Defense Strategy Design. The proposed defense mechanism largely relies on cosine similarity and Euclidean distance to filter and weight updates. However, cosine similarity–based direction checks are already widely used in existing Byzantine-robust aggregation schemes (e.g., FLTrust, BALANCE, and various local similarity-based defenses). The current paper does not provide a convincing argument for why using cosine similarity within MPC constitutes a novel or more effective approach.

2. Low efficiency. The paper claims to implement decentralized federated learning, yet assumes a fully connected peer-to-peer network(Client i need share its local weight to all of the other clients). This assumption contradicts the practical reality of decentralized federated learning and leads to enormous communication complexity.

3. Use of MAC-Based Secret Sharing Is Standard. The cryptographic layer builds directly on Cramer et al. (2018)’s authenticated additive secret sharing (MAC-based MPC). While this is a sound and practical choice, it is a well-established protocol and the paper does not introduce any modification or optimization to it. Therefore, the claim that the privacy layer is a key innovation seems overstated — the novelty lies more in system integration rather than cryptographic or algorithmic advancement. By the way, the equation in line 161(the last line in page3) is uncorrect.

4. Positioning Relative to Existing Work. The comparison to Franzese et al. (NeurIPS’23), BALANCE (CCS’24), and other decentralized secure frameworks could be more critical. Many prior works already combine secret sharing or MPC backends with robust aggregation. The current contribution may thus be incremental unless the authors can demonstrate measurable privacy–robustness improvements beyond reimplementation.

**Questions:**

see the above

---

### Official Review · Reviewer_eqEc · 2025-11-01

**Soundness:** 2
**Presentation:** 3
**Contribution:** 2
**Rating:** 2
**Confidence:** 4

**Summary:**

This paper addresses the fundamental conflict between robustness and privacy in fully decentralized learning (DL). In DL, clients must access each other's model updates to perform robust aggregation, but this exchange makes them vulnerable to privacy-inference attacks.
The authors introduce SecureDL, a protocol described as the first to provide both Byzantine resilience and strong privacy guarantees in this decentralized setting.

SecureDL achieves this by using secret sharing-based multi-party computation (MPC). This allows clients to collaboratively run a robust aggregation algorithm on their encrypted model shares without revealing their raw models to each other . This secure process involves a cosine-similarity filter and uses a scoring-based system to aggregate the trusted, updates.

The authors claim that SecureDL can guarantee privacy-preserving, robust training even in a dishonest majority setting. Extensive evaluations show that the protocol maintains high accuracy and outperforms other decentralized defenses against a wide range of data and model poisoning attacks.

**Strengths:**

- It addresses the fundamental tension in decentralized learning: the conflict between Byzantine-robustness and privacy. This has been acknowledged in literature before but no single system has yet resolved this conflict. The paper claims SecureDL is the first protocol to solve this.
- The protocol makes a strong security claim, stating it requires only two semi-honest and non-colluding clients to guarantee privacy-preserving, robust training, regardless of the total number of Byzantine participants.
- SecureDL allows for weighted averaging post cosine filtering in the encrypted domain all without revealing the individual models using secret sharing.
- The paper includes a formal proof (Theorem 2) to theoretically guarantee its privacy-preserving properties against malicious adversaries.

**Weaknesses:**

- From Algorithm 1, it appears every node breaks its model into shares and distributes one to each neighbor. For a node to decrypt the aggregated model, it seems to imply that all its neighbors must be fully connected. It is unclear how this is coordinated in the experiment simulations, especially if the graph is not static. This is a disadvantage, as it does not allow a node to independently choose its neighbors.
- The experiments were done with only 10-30 clients, which is too small. With such a low number of clients, the effect of non-IIDness cannot be studied properly. This small scale fails to capture the core difficulty of distinguishing between malicious models and benign models trained on unique data - the very problem that makes securing private, decentralized learning so difficult.
- The results showing SecureDL survives attacks better than others may be primarily because the aggregation design encourages "echo chambers." Each client uses its own model as the reference, filtering out most dissimilar updates. This effect would have been more prominent with more clients, but the primary experiments used only 10 nodes. The paper also lacks any analysis of false positive or false negative detection rates.
- The attacks used, such as standard Sign-Flipping and Gaussian Noise, are relatively weak. Stronger, more optimized versions of these attacks could have been used. The paper must also survey important recent defense baselines.
-In decentralized learning, all models can be distinct, making cosine distance between two models (rather than gradients) an unreliable metric for similarity. This metric makes sense in FL where gradients are compared against a single reference model, but when all models are distinct, it cannot be directly used. A more robust approach would be to compare the direction in which the models are moving.
- The discussed results cannot be generalized because the defense is heavily dependent on a cosine similarity threshold. The paper lacks a thorough study on the effect of this threshold.
- Even with only 30 clients, the overhead mentioned in Table 4 is huge. This, combined with the $O(n^3)$ communication complexity, makes the design unscalable for practical use.

**Questions:**

Do the nodes form a hard clustered graph where every cluster is fully connected internally but has no connections with any other nodes outside the cluster?

---

### Note · Authors · 2025-11-22

**Comment:**

Thank you for the time and effort the reviewers and committee invested in evaluating our submission. After consideration, we have decided to withdraw the paper. We appreciate the constructive feedback and your understanding.

**Withdrawal Confirmation:**

I have read and agree with the venue's withdrawal policy on behalf of myself and my co-authors.